

# Plant community composition controls spatial variation in year-round methane fluxes in a boreal rich fen

Eeva Järvi-Laturi[1], Teemu Tahvanainen[2], Eero Koskinen[3], Efrén López-Blanco[4,5], Juho Lämsä[3], Hannu Marttila[1], Mikhail Mastepanov[3,4], Riku Paavola[3], Maria Väisänen[6], Torben R. Christensen[1,4]

[1]Water, Energy and Environmental Engineering Research Unit, University of Oulu, Oulu, 90570, Finland
[2]Department of Environmental and Biological Sciences, University of Eastern Finland, Joensuu, 80100, Finland
[3]Oulanka Research Station, University of Oulu, Kuusamo, 93900, Finland
[4]Department of Ecoscience – Arctic Ecosystem Ecology, Aarhus University, Roskilde, 4000, Denmark
[5]Department of Environment and minerals, Greenland Institute of Natural Resources, Nuuk, 3900, Greenland
[6]Ecology and Genetics Research Unit, University of Oulu, Oulu, 90570, Finland

*Correspondence to*: Eeva Järvi-Laturi (eeva.jarvi-laturi@oulu.fi)

**Abstract** Climate change is expected to impact the methane budget of boreal peatlands, highlighting the need to understand the factors that influence methane cycling, including plant community structure. In northern peatlands, the majority of methane is transported through plants, and the magnitude of this process is strongly linked to plant community composition. Therefore, detailed information about the role of plants regulating year-round methane fluxes is highly valuable. This paper explores the causes of spatial variability in plot-scale methane fluxes in a northern boreal rich fen. Methane fluxes were measured using the manual chamber technique in the context of fine-scale biomass variations in plant community compositions from 36 study plots over 232 days throughout a full year. The mean methane flux rates for snow-free and snow seasons were 2.55 and 0.21 mg $CH_4$/m$^2$/h, respectively. We found a significant correlation between methane fluxes and a vascular plant cluster associated with the occurrence of the sedge *Carex rostrata* during year-round, snow-free and snow season periods. More precisely, *C. rostrata* grew at the point of flux measurement in 13 plots and 44–49 % of the measured methane fluxes originated from these plots during the three periods. The biomass of vascular plants, sedges, and *C. rostrata*, as well as the ratio of vascular plant to bryophyte biomass, also significantly correlated with methane fluxes in year-round and snow-free season. By identifying vegetation-driven emission hotspots, these results can enhance efforts to upscale emission predictions and improve ecosystem-scale methane modelling. Thus, our findings provide valuable insights for predicting realistic future changes in peatland methane emissions throughout the year.

## 1 Introduction

Northern peatlands are an intrinsic part of the global carbon cycle and currently, these peatlands store more than a third of all terrestrial carbon, act as strong sinks of carbon dioxide ($CO_2$) and are the main natural terrestrial source of methane (Ramage et al., 2024; Schuur et al., 2022). Indeed, natural wetlands produce 22–30 % of the global methane emissions that are still



considered an important source of uncertainty in the global methane budget (Saunois et al., 2020). The uncertainty arises from several factors: the relative contributions of methane emissions from tropical and northern wetlands, how these regions respond to rising temperatures, and the spatial and temporal dynamics of the emissions (Christensen, 2024; Yuan et al., 2024).

Climate change is predicted to affect the hydrology of peatlands by increasing the water table depth (WTD) (Evans et al., 2021; Helbig et al, 2020; Swindles et al., 2019), which is one of the most well-known regulators of methane dynamics along with temperature and vegetation (Turetsky et al., 2014). According to several studies, an increased WTD would decrease methane fluxes (e.g. Pearson et al., 2015; Riutta et al., 2020) and increase the rate of decomposition and soil $CO_2$ emissions (Ma et al., 2022). This ecosystem process is complex, though, as the increasing level of atmospheric $CO_2$ is predicted to
enhance plant productivity (Forkel et al., 2016) and thereby the rate of root exudation (Nielsen et al., 2017), leading to greater methane emissions (Turner et al., 2020). The expected rise in methane production could be balanced by increased methane oxidation in the topsoil layers, which is a probable response to rising temperatures (Zhang et al. 2021). Warming climate may, however, also increase the areal cover of wet fens in the Arctic region due to permafrost thaw, which could potentially create new sources for methane release (e.g. Christensen et al., 2023; Grimes et al., 2024).

In addition to hydrology, vegetation type and responses to environmental changes are highly relevant for methane dynamics, as up to 90 % of ecosystem-level methane in northern peatlands is transported through plants (Ge et al., 2023; Korrensalo et al., 2022). The aerenchymatous tissues of certain vascular plant species allow methane to move from deeper soil through the plant, thus avoiding the oxidation in upper soil layers (Ge et al., 2023; Joabsson et al., 1999). Indeed, plant species and their specific traits predict methane flux rates better than any studied abiotic factor (Korrensalo et al., 2022). *Sphagnum*
mosses of wet environments can also host methanotrophic microbes and thus have a potential to oxidate methane and affect the magnitude of the total emissions (Larmola et al., 2010). Climate change is predicted to accelerate the natural vegetational succession in boreal rich fens towards *Sphagnum*-dominated plant communities even in stable hydrological conditions (Kolari et al., 2021), which could have major impacts on methane dynamics. To improve our understanding and the predictions of future methane emissions, it is important to have more focus on the vegetation composition and the specific plant species
controlling the magnitude of the fluxes (Riutta et al., 2020).

The relationship between plant community composition and methane fluxes remains an important topic of study (e.g. Lai et al., 2014; Riutta et al., 2007; Ström et al., 2015) and research on individual plant species has shown significant variation in the magnitude of flux rates and transport efficiencies (e.g. Bhullar et al., 2013; Koelbener et al., 2010; Korrensalo et al., 2022). Studies from northern boreal rich fens with extensive, year-round, plot-scale flux data are, however, limited. This study
aims to better understand the causes behind local spatial differences of methane fluxes, and to provide a new perspective on assessing plant-mediated methane emissions. To this end, we focus on fine-scale variations in plant community composition based on species' biomass, using non-destructive in situ methods. We intend to answer the following questions: (1) Do methane flux variations correlate with plant community type at a study plot scale? (2) Does plant community composition correlate with methane flux variability alone or in combination with other environmental factors? We hypothesize that (1) the plant





community composition affects the methane flux and that (2) the flux is highest on study plots with largest biomass of vascular
plants in absolute terms or in proportion to the biomass of bryophytes.

## 2 Materials and methods

### 2.1 Study site

This study was implemented in Puukkosuo, an open and slightly sloping calcareous fen located in the northern boreal zone at
Oulanka National Park in Kuusamo, Northeast Finland (66.377299° N, 29.308062° E) (Fig. 1). The mean annual, January and
July temperatures from the normal period of 1992–2022 were 0.6, -13.0, and 15.3 °C, respectively, and the mean annual
precipitation was 557.4 mm. The study period (19.10.2021–31.10.2022) was slightly warmer and drier than the normal period
with temperatures 0.9, -11.4, and 16.5 °C, respectively and with total annual precipitation of 528.8 mm. The mean pH during
the snow-free season of 2022 was 7.0, ranging from 6.74 to 7.38.  The deepest measured water table was 9.3 cm below the
peat surface, while the highest was 7.0 cm above the surface (Fig. A1). The WTD fluctuated only slightly: during the snow-
free season of 2022 the variation in WTD throughout the fen was 6.3 cm on average. The vegetation is dominated by
graminoids (*Carex* sp., *Trichophorum* sp., *Molinia caerulea*), herbaceous plants (*Potentilla erecta*, *Menyanthes trifoliata*), as
well as brown mosses (*Scorpidium cossonii, Campylium stellatum, Cinclidium stygiym*) and peat mosses (*Sphagnum* spp.,
mostly *S. warnstorfii*).

### 2.2 Experimental design

The study area was approximately one hectare in size and included 12 spatial blocks. Each block had three study plots resulting
in 36 study plots. The plots were established in summer 2018, and the size of a plot was 2 m × 3.5 m which included a 0.5 m
wide buffer zone. Wooden boardwalks, built to minimize stepping on the surface of the peat, led to the plots. Half of the study
plots (n=18) were located inside a fence, built in spring 2019, to exclude grazing by reindeer (Fig. 1). At the time of this study,
the exclusion had lasted for 2–3 years. The location of the plots and the fence followed the hydrological gradient of the fen
(Fig. 1). The study plots were also assigned to snow level manipulations that were started in January 2019. Within each block,
one plot was an untreated control with ambient snow level, one a snow removal plot where the snow depth was maintained at
0.25 m throughout the snow season, and one plot a snow addition plot, where the snow from the removal plot was placed. The
snow treatments had no statistically significant effects on the methane fluxes (Fig. B6).



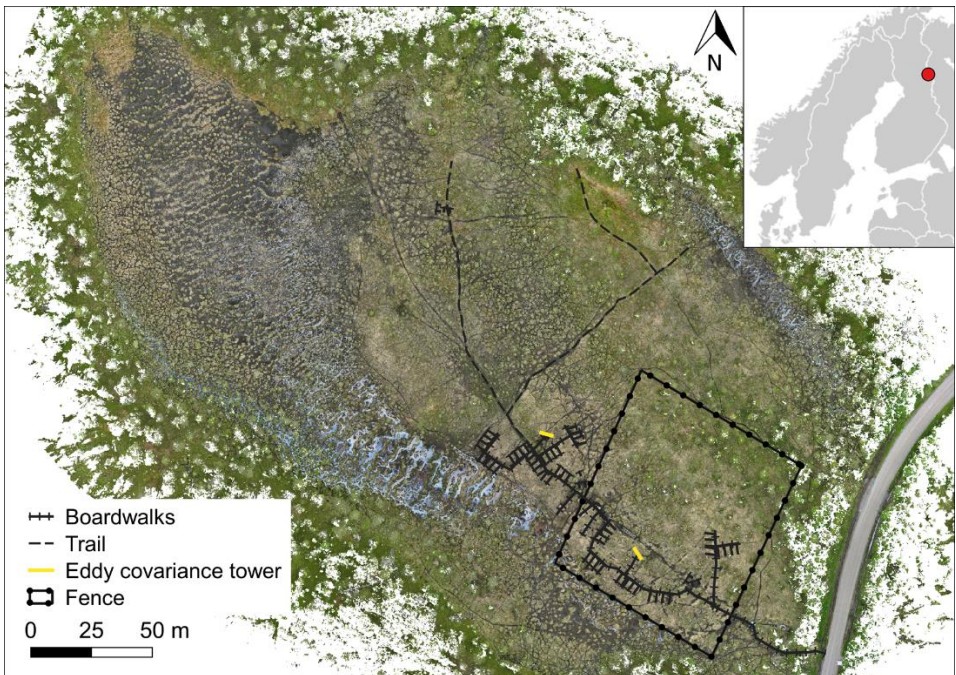


**Figure 1. A map of the study system, Puukkosuo rich fen showing the location of boardwalks, experimental area and the fence excluding reindeer. The small inset shows the location of the study area, Oulanka, in NE Finland. Orthomosaic © Petra Korhonen 2024.**

### 2.3 Methane flux measurements

For manual methane flux measurements, a round PVC collar (inner diameter 29 cm) was inserted approximately 5 cm into the ground in the rear end of each plot in September 2020. During 19.10.2021–31.10.2022, we measured methane fluxes (mg $CH_4/m^2/h$) over 232 individual days between 8 am and 6 pm, doing measurements from half (n=18) of the study plots per day. We used manual, closed chamber technique (e.g. Christensen et al., 2000) with a portable LI-COR $CH_4/CO_2/H_2O$ Trace Gas Analyzer LI-7810 and a transparent Plexiglas chamber (height 38 cm, diameter 29 cm) equipped with a small fan to circulate

the air inside the chamber (Fig. 2a). For each measurement during snow-free season, the chamber was placed on the PVC collar for an airtight seal (Fig. 2b). Each measurement lasted for 5 minutes. During winter, when snow covered the collars and it was not possible to place the chamber directly on them, we measured the fluxes on top of the snowpack, also known as floating chamber technique (Björkman et al., 2010). These measurements were taken at a slightly different spot closer to the boardwalks to avoid unnecessary disturbance to the snowpack. Clearing the measuring points from snow would have caused

excessive disturbance to both the snowpack and the continuous flux measurements, making the possible dilution of the flux during diffusion through the snowpack inevitable (Björkman et al., 2010). Due to these limitations, snow season flux measurements should be examined with caution. Altogether, 4319 individual measurements were taken, of which 4121 were successful and used in this study.



To calculate methane flux rates for each plot, we used a Python script that calculated the slope of methane concentration change during three centremost minutes of a five-minute measurement period and computed the flux in mg $CH_4/m^2/h$ using ambient air pressure and air temperature at the time of each measurement (linear regression model, e.g. Pirk et al., 2017). We accepted the measurements with an $R^2$ value $\geq 0.95$ (n = 3589) and inspected all the rest (n = 691) individually, leaving out measurements showing very strong non-linearity or any other sign of failed measurement (n = 159). We examined the fluxes in three periods: 1) year-round (19.10.2021–31.10.2022), 2) snow-free (13.5.–26.10.2022), and 3) snow season (19.10.2021–12.5.2022 and 27.10.–31.10.2022). Snow-free and snow seasons were defined by the snow cover and the ability to measure the fluxes on the collar. Annual accumulated flux was estimated by calculating a 24-hour accumulated flux for each available datapoint by multiplying the hourly mean flux by 24. These daily flux values were then summed to obtain the annual total. The days which were missing a measurement were given the value from a previous measurement. In the annual accumulated flux calculations, we assumed that the fluxes did not vary remarkably diurnally or over the days.

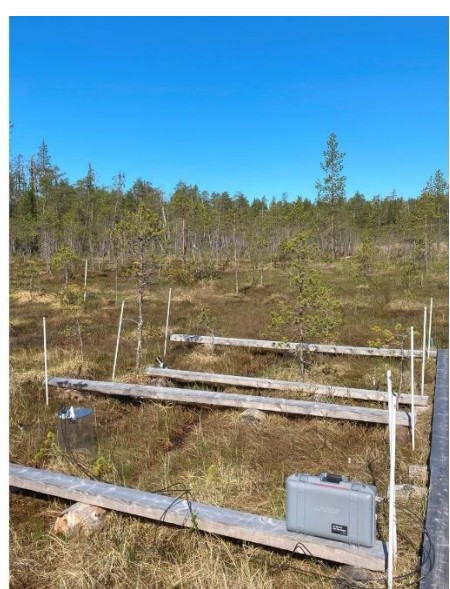
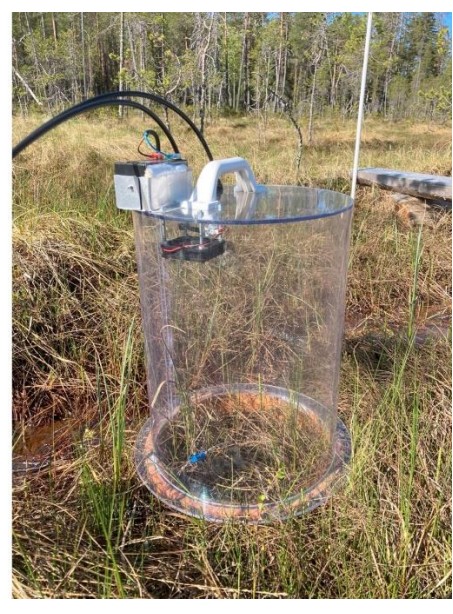

**Figure 2. A flux measurement carried out with (a) a portable LI-COR Trace Gas Analyzer and (b) a clear plexiglas chamber placed on a round collar installed at each plot in 2020. Photos by Eeva Järvi-Laturi.**

**2.4 Plant community data**

We studied the plant communities by identifying the species within each collar and estimating their biomasses (BM) from separately collected samples during 25.7.–12.8.2022. To survey the vascular plant (VP) species (n=31) within each collar, we counted each aboveground shoot individually and separated them into fertile and sterile categories. Then we collected separate samples (n=10 for both fertile and sterile shoots) of each identified species for BM estimation by cutting the shoot at peat surface. For bryophytes, we visually estimated the percentage coverage of each species (n=10) within each collar. We took





three samples covering 5 % of the collar area for the five most common species, and one sample covering 1 % of the area for
the remaining five species, then cut them to represent the aboveground biomass. All BM samples were collected outside of the
main research area nearby the study plots. We dried the samples (n=454) in a hot air circulation oven at 40 °C for minimum
of two days and weighed them in a laboratory using a Denver Instrument SI-234 analytical scale with a four decimal precision
(Table A1). We acknowledge that studying the plant communities using non-destructive methods and a mean BM of the
separate samples, instead of the actual BM of the plant communities, introduces a margin of error in the calculations.

### 2.5 Environmental variables

We used weekly measures of WTD from June to October from regular measuring points located approximately 1 m from the
methane flux measurement points. At every plot, soil temperature was recorded with 10-min intervals at 5 cm depth (model
p107 CS CR1000, Campbell Scientific Inc., Logan, UT, USA).  In October 2023, we measured peat layer thickness at the rear
edge of each study plot with a thin, metallic 300 cm long auger. To determine peat chemistry, we collected pore-water during
frost-free periods. In the beginning of snow-free season 2022, rhizon samplers (Rhizosphere Research Products, The
Netherlands) were installed (10 cm depth) for pore water sampling in the middle of the collar in each experimental plot. Pore
water was sampled five times during the frost-free periods (31 May, 29 June, 22 July, 28 August, 29 September). Samples
were collected into evacuated opaque syringes over a period of 24–48 h, filtered (0.45 µm, sterile nylon, Sarstedt, Germany)
and frozen (−18 °C). Thawed samples from all sampling campaigns were analyzed for pH (Metrohm), dissolved organic and
inorganic carbon (DOC, DIC; Shimadzu DOC-VCX, Trios) and dissolved organic nitrogen (DON), ammonium (NH4) and
nitrite + nitrate (NO2+NO3; AA500 Seal Analytical) and for DOC and DON reported as mg/l and for NH4 and NO2+NO3 as
µg/l. Additionally, we observed the amount of litter inside the collar (%) and the microtopography around the collar
(flark/intermediate/hummock) while identifying the vegetation.

### 2.6 Data analyses

To analyse the vegetation, we divided the plant communities into three species combinations: 1) all species, 2) VPs, and 3)
bryophytes. We analysed these combinations separately by hierarchical cluster analysis using Sorensen (Bray–Curtis) distance
measure and Flexible Beta group linkage method (McCune et al., 2002) with beta-value of -0.25. Clusters with less than six
samples (study plots) were discarded. To evaluate which species were statistically most connected to the different clusters, we
carried out an indicator species analysis separately to all three species combinations. We tested the differences between the
clusters with Tukey's HSD (Honestly Significant Difference) test and the clusters' relation to methane fluxes in different time
periods (snow-free, year-round and snow season) with linear regression models using the lm function (R Core Team, 2024).
We performed detrended correspondence analysis (DCA) to identify patterns in species composition and canonical
correspondence analysis (CCA) to relate the species composition to environmental variables (McCune et al., 2002). We
conducted the analyses separately for VPs and bryophytes to determine the main environmental factors characterizing the



composition of these plant communities. Finally, we conducted local regression models (LOESS, locally estimated scatterplot smoothing) to study the significance between BM variables and methane fluxes in relation to VP clusters.

We estimated the plot-scale aboveground BM of each plant species using a mean dry mass from separate samples (Table A1). We also calculated a VP to bryophyte ratio (%) for each plot (Table A2) and analyzed the relationships of multiple

BM and environmental variables with a correlation matrix (Table B2). Cluster, indicator species and correspondence analyses were implemented with PC-ORD version 7.09 (McCune and Mefford, 2018). Regression models and Tukey's HSD tests were performed using RStudio version 2024.4.2.764 (Posit team, 2024). Both programmes were used for data visualization: PC-ORD for cluster dendrograms, boxplots, and ordination graphs, RStudio for line graphs and scatterplots. We used the following R packages: readxl v1.4.3 (Wickham and Bryan, 2023), dplyr v1.1.4 (Wickham et al., 2023a), tidyr v1.3.1 (Wickham et al.,

2024), ggplot2 v3.5.1 (Wickham, 2024), forcats v1.0.0 (Wickham, 2023), scales v1.3.0 (Wickham et al., 2023b), paletteer v1.3.0 (Hvitfeldt, 2021), ggnewscale v0.5.0 (Campitelli, 2024) and viridis v0.6.5 (Garnier et al., 2024). R scripts were created with the assistance of Microsoft 365 Copilot, an AI-powered productivity tool.

## 3 Results

### 3.1 Methane fluxes

Methane fluxes were the highest during the snow-free season with an overall mean of 2.55 mg $CH_4/m^2/h$. The variation in methane fluxes across all study plots was also greatest during this period, with the mean of plot-scale fluxes varying between 0.51 and 4.67 mg $CH4/m^2/h$. The highest individual fluxes per plot varied between 1.51 and 9.17 mg $CH_4/m^2/h$. The lowest flux values of the snow-free period (0.02–1.18 mg $CH_4/m^2/h$) were measured in May after a spring burst (1.4.–12.5.2022), and at the end of the season in late October (Fig. 3). The magnitude of the spring burst differed among the plots, with the maximum

individual fluxes ranging from 0.15 to 6.65 mg $CH_4/m^2/h$. During the snow season, the overall mean flux was 0.21 mg $CH_4/m^2/h$, with the mean of plot-scale fluxes ranging from 0.07 to 0.56 mg $CH4/m^2/h$. On a year-round scale, the overall mean flux was 1.37 mg $CH_4/m^2/h$, with mean fluxes varying between 0.29 and 2.52 mg $CH_4/m^2/h$.



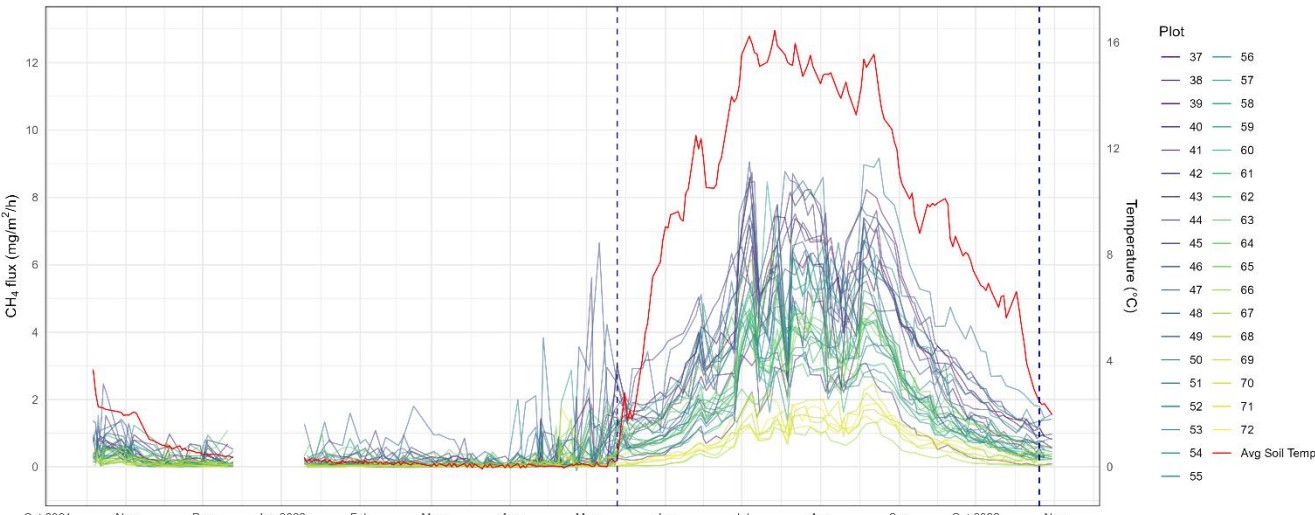

**Figure 3. Year-round plot-scale methane flux (mg CH$_4$/m$^2$/h) and mean soil temperature at 5 cm depth from the end of October 2021 to the end of October 2022. The red line represents the average soil temperature of all plots, while the other lines represent a manually measured flux of each individual study plot. The vertical dashed lines mark the start and end of the snow-free season (13.5.– 26.10.2022).**

### 3.2 Plant community structures

The plots differed in their plant community structures, with wide variation in plant BM. VP BM per plot ranged from 62.1 g/m$^2$ to 486 g/m$^2$, while the bryophyte BM varied between 65.1 g/m$^2$ and 269 g/m$^2$. The lowest total BM for an individual plot was 167 g/m$^2$, while the highest was 674 g/m$^2$. The percentage of VPs in the total BM varied between 21 % and 72 % (Table A2). Species producing the most BM were sedges *Carex lasiocarpa* (122 g) and *C. rostrata* (84 g), and bryophytes *Scorpidium cossonii* (193 g), *Campylium stellatum* (113 g), and *Sphagnum* spp. (92 g) (Table A3).

The cluster and indicator species analyses for all species, VPs, and bryophytes all yielded three clusters (groups of plots) with significant (p < 0.05) indicator species statistically connected to the clusters (Figs. B1–B3). The cluster analyses indicated that bryophytes (*Sphagnum* spp.*, S. cossonii*, and *C. stellatum*) showed the strongest connection to the clusters when analyses were done with all plant species or bryophytes (Figs. B1, B3). On the other hand, when analysing VPs alone, different sedges connected with the clusters [*C. rostrata* (C.ros), *C. chordorrhiza* (C.cho), and *Trichophorum cespitosum, C. lasiocarpa*, and *Potentilla erecta* (T.ces)]. The characteristics of each VP cluster, including their community structure and indicator species, were studied by comparing the indicator values of the VP species (Table B1). The results demonstrated that the plant communities differed between the clusters, as most species were abundant in only one or two clusters.




### 3.3 Clusters' relation to methane fluxes

VP clusters correlated significantly with snow-free season (F = 10.71, p < 0.001) and year-round (F = 10.92, p < 0.001)
methane fluxes but not with snow season fluxes (F = 2.14, p > 0.05). Here and in the following, significance is defined as p <
0.05. The C.ros-cluster, which had the highest fluxes especially during snow-free season (Fig. 4), differed significantly from
the C.Cho- and T.Ces-clusters in snow-free (p < 0.01 and p < 0.001, respectively) and year-round periods (p < 0.01 and p <
0.001, respectively) but not in snow season (Fig. 5a). There were no significant differences between C.cho- and T.ces-clusters
in any of the periods. All species-clusters did not correlate significantly with methane fluxes in any period (F = 1.51, p > 0.2
for snow-free, F = 1.45, p > 0.2 for year-round, and F = 0.57, p > 0.5 for snow season). Similarly, the bryophyte clusters did
not correlate significantly with methane fluxes (F = 1.23, p > 0.3 for snow-free, F = 1.26, p > 0.2 for year-round, and F = 0.90,
p > 0.4 for snow season) (Figs. 5b, c).

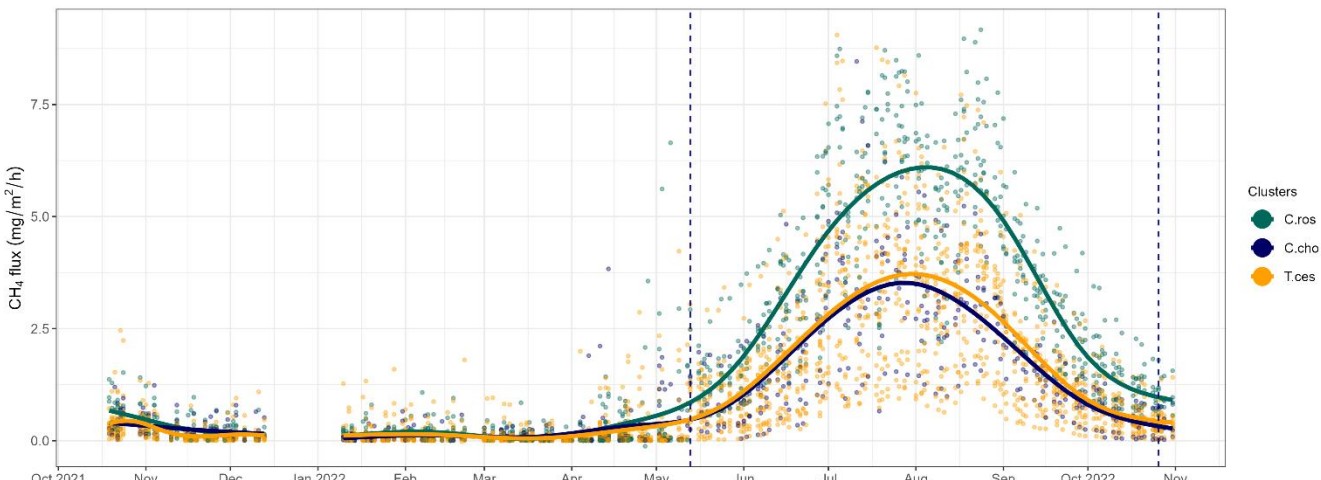

**Figure 4. Year-round methane fluxes of Puukkosuo with smoothed conditional means segregated by vascular plant clusters. See Fig.**
**5 for cluster abbreviations. Each dot in the graph represents one manual flux measurement. The vertical dashed lines mark the start**
**and end of the snow-free season (13.5.–26.10.2022).**





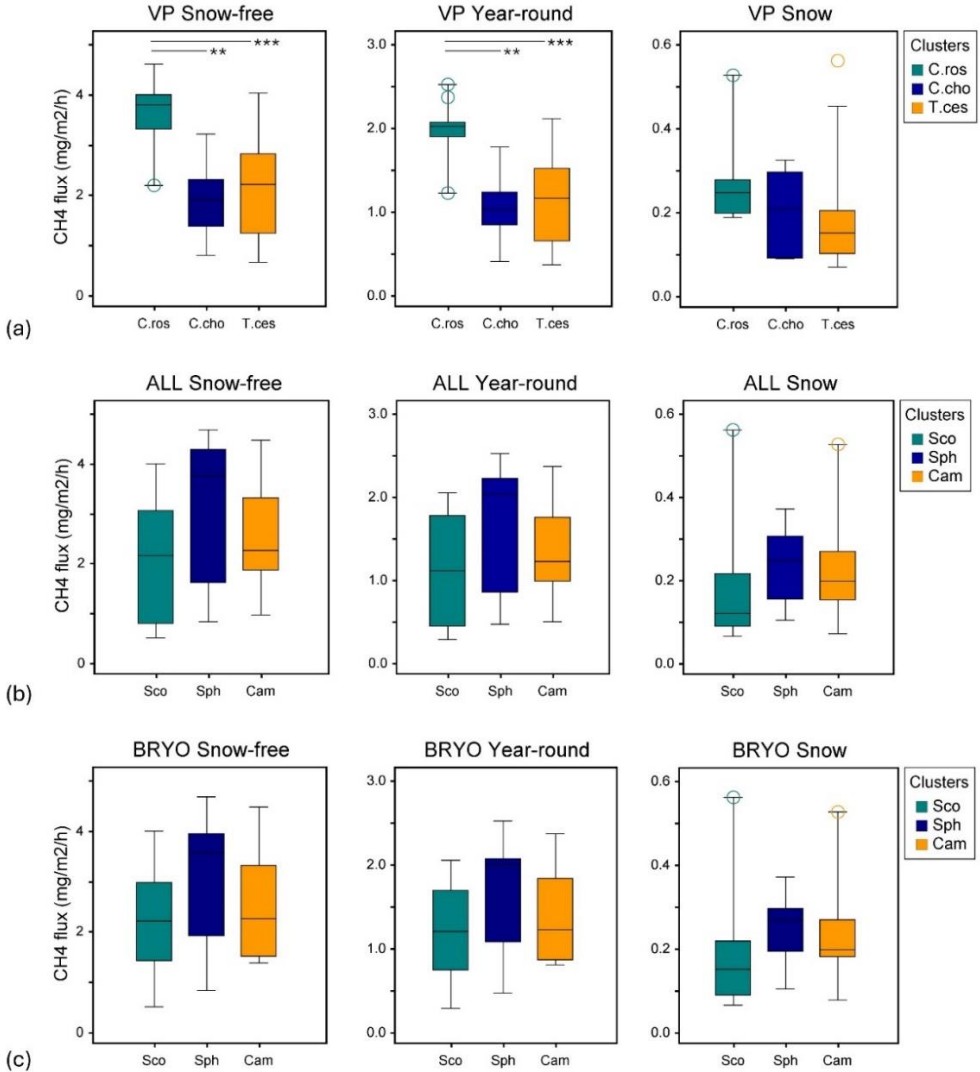

**Figure 5. Methane fluxes in snow-free season (left panel), the whole year (middle panel), and snow season (right panel), as divided**
**by cluster analyses of (a) vascular plants (VP), (b) all species (ALL) and (c) bryophytes (BRYO). Abbreviations for the clusters and the species with highest fidelity according to indicator species analysis: C.Ros = *C. rostrata*, C.Cho = *C. chordorrhiza*, T.Ces = *C. lasiocarpa*, *P. erecta*, and *T. cespitosum*, Sco = *S. cossonii*, Sph = *Sphagnum* spp. and Cam = *C. stellatum*. Asterisks above the bars denote significant differences between the clusters at the level p < 0.01\*\* and p < 0.001 \*\*\*.**

### 3.4 Ordination analyses

Within VP communities, the C. ros-cluster diverged from the C.cho- and T.ces-clusters in both DCA and CCA, while the C.Cho- and T.Ces-clusters showed a clear distinction in DCA, but not in CCA (Fig. B4a–b). Similarly, in bryophyte communities the Sph-cluster diverged from the Sco- and Cam- clusters in both DCA and CCA, while the Sco- and Cam-clusters showed a clear distinction in DCA, but not in CCA (Fig. B5a–b). The main compositional gradients of VP and




bryophyte communities displayed different correlation patterns with environmental variables. The first ordination axis of the
VP communities correlated with snow-free and year-round methane fluxes (r = 0.775 and 0.782 in DCA, r = -0.856 and -0.866
in CCA, respectively, Fig. B4a–b). Additionally, the first ordination axis correlated with snow season fluxes (r = 0.445 in
DCA, -0.402 in CCA). The correlation was also significant for the ratio of VP and bryophyte BM (r = 0.562 in DCA, -0.730
in CCA), peat layer depth (r = 0.342 in DCA and -0.580 in CCA) as well as the combined nitrate ($NO_3^-$) and nitrite ($NO_2^-$)
concentration (r = 0.529 in DCA, -0.453 in CCA). In contrast, the strongest compositional gradient of the bryophyte data
correlated with WTD and pH in both DCA (r = -0.606 and 0.473, respectively) and CCA (r = -0.614 and 0.408, respectively)
(Fig. B5a–b). The correlation between the bryophyte data and methane was not significant in any of the periods (r = 0.267,
0.262, and 0.132 in DCA, r = 0.316, 0.326, and 0.130 in CCA in snow-free, year-round, and snow season, respectively).

### 3.5 BM, environmental variables and methane

The total BM of VPs correlated with the total BM of sedges (r = 0.98) and the total BM of *C. rostrata* (r = 0.93) indicating
that sedges were the main functional group of VPs, and *C. rostrata* the main VP species, producing BM in our study site. The
BM variables that had a significant correlation with methane fluxes in year-round and snow-free season were the total BM of
VPs (p < 0.001) (Fig. 6a), the total BM of sedges (p < 0.001), the total BM of *C. rostrata* (p < 0.001) (Fig. 7), and the ratio
between VP and bryophyte biomasses (from here called the BM ratio) (p < 0.01) (Fig. 6b). Significant correlations during the
snow season were not discovered. Environmental variables that had a significant correlation with methane fluxes in snow-free
and year-round periods were pH (p < 0.05 and ≤ 0.01, respectively) and combined concentration of $NO_3^-$ and $NO_2^-$ in peat pore
water (p < 0.05 for both periods). There was no significant correlation between methane fluxes and WTD or soil temperature
in any period. All correlation coefficients are listed in a correlation matrix in Table B2.

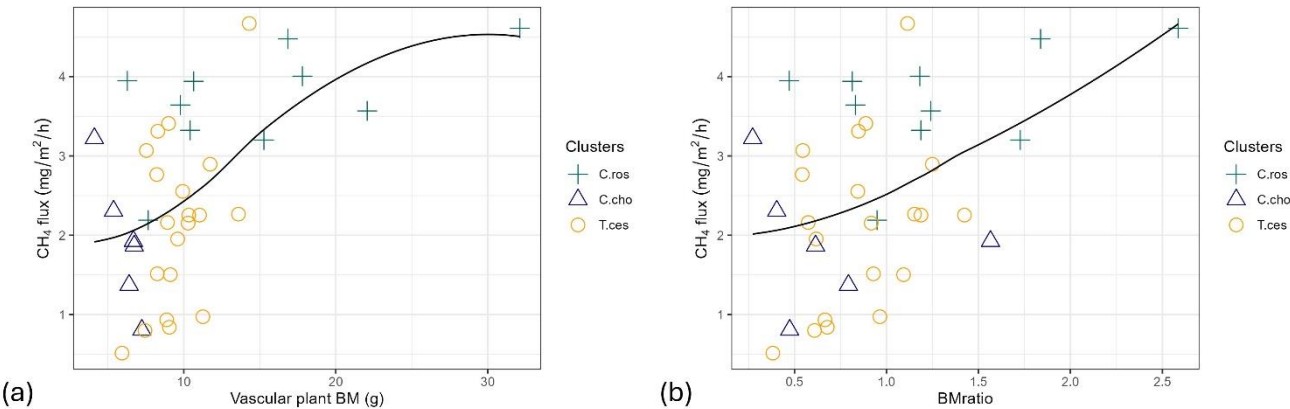

**Figure 6. The relation of snow-free season methane fluxes (mg/m²/h) and (a) total biomass (BM) of vascular plants (g of dry weight) (p < 0.001, r = 0.54) and (b) biomass ratio of vascular plants and bryophytes (p < 0.01, r = 0.45) segregated by vascular plant clusters. See Fig. 5 for cluster abbreviations.**



## 4 Discussion

### 4.1 Methane flux variation over time and space

During the snow-free season, the range of fluxes was 0.02–9.17 mg $CH_4/m^2/h$ (i.e., 0.48–220 mg $CH_4/m^2/d$), with the overall mean being 2.55 mg $CH_4/m^2/h$ (i.e., 61 mg $CH_4/m^2/d$). Similar flux rates ranging from ~30–300 mg $CH_4/m^2/d$, with a mean of ~100 mg $CH_4/m^2/d$ during June–October (Jammet et al., 2017) have been measured from northern boreal rich fens. The snow season fluxes' range was 0–6.65 mg $CH_4/m^2/h$ (i.e., 0–160 mg $CH_4/m^2/d$), with an overall mean being 0.21 mg $CH_4/m^2/h$ (i.e., 5 mg $CH_4/m^2/d$). Further, the snow season fluxes accounted on average for 8.2 % of the estimated annual accumulated flux,

with values ranging from 2.3 % up to 21.3 % across the study plots (Table B3). Even though we measured fluxes on top of an undisturbed snowpack – which may underestimate the magnitude of the fluxes to some extent (e.g. Björkman et al., 2010) – our results are in line with a study from boreal fens in central Finland, where 6–17 % of the annual methane release was observed during wintertime, when measuring the fluxes on peat surface after clearing the spots from snow (Alm et al., 1999). Given that winter may account for up to 20 % of the annual methane flux in boreal fens, any changes in wintertime processes

may impact future methane emissions from these regions. Our results highlight the importance of including winter in methane flux studies, as this information may help to reduce the current large uncertainties in the net carbon balance.

Flux rates during the snow-free season reached up to 9.17 mg $CH_4/m^2/h$ in study plots 45 and 50, while they never exceeded 2 mg $CH_4/m^2/h$ in study plots 67, 68, 71 and 72 (Fig. 3). The high fluxes in plot 50 may partly be explained by the high biomass of vascular plants and *Carex rostrata*. Indeed, the presence of *C. rostrata* appears generally important for the

spatial variability in fluxes: all plots with more than five flux measurements exceeding 6 mg $CH_4/m^2/h$ (n = 9) contained *C. rostrata* shoots. In contrast, only three out of 19 plots that never recorded fluxes above 6 mg $CH_4/m^2/h$ contained *C. rostrata*. However, *C. rostrata* biomass cannot be the sole factor of high methane fluxes, as plot 45 did not contain any *C. rostrata* shoots. Unlike the snow-free season, the high flux values from plots 43, 46, 47, 49 during the spring burst at the end of the snow season could not be explained by vegetation or any other studied environmental variable. Other reasons for the spatial

variation in flux rates may include species-specific plant traits (Ge et al., 2023), ecohydrological aspects (Zhang et al., 2020), or the role of microbiota (Kujala et al., 2024; Yavitt et al., 2012). Interestingly, the methane fluxes from *C. rostrata*-community plots were the highest from late July to late August (Fig. 4) when vegetation at our site remained predominantly green, even though fluxes from *C. rostrata* shoots have been reported to be the highest in early autumn when leaves are senescing (Ge et al., 2024). The period from late July to late August may coincide with the peak development of permeable root surface area

and aerenchyma (Reid et al., 2015). The extent of permeable root surface, in turn, is a key factor influencing methane transport in plants (Henneberg et al., 2012). Therefore, the seasonal changes in methane flux rates associated with *C. rostrata* may be controlled by the belowground parts of the plant (Ge et al., 2024).



## 4.2 Plant communities and methane

While previous studies have shown substantial vegetation–methane flux correlations during the growing season (e.g. Lai et
al., 2014; Riutta et al., 2007; Ström et al., 2015), our research extends this to year-round scale, finding significant correlations
in all studied time periods; snow-free, year-round and snow season, and thus supporting our first hypothesis. We acknowledge
that the measured snow season fluxes might not fully represent the identified plant communities, as the measurements were
not taken straight above the collars and the flux rates from different parts of the plots may differ to an unknown extent.
However, our results indicate that the influence of plant communities on methane flux dynamics is not limited to active growing
season or plant senescence, and that some species are likely to be more efficient at supporting methane production and
transport, even under the snowpack.

The differences in plant community structures between the study plots, in relation to methane fluxes, were discovered
when the focus of cluster and indicator species analyses was set solely on vascular plants (Fig. 5a). The clusters were mostly
associated with different sedges and had only three species (*Campylium stellatum*, *Vaccinium oxycoccos*, and *Scorpidium
cossonii*) common to all clusters (Table B1). At the peatland level, the spatial division of the vascular plant communities was
clear – the *C. rostrata*-community plots were all located in the upper part of the sloping fen, while the other two clusters were
more widely distributed (Fig. B7). Edaphic factors, such as water table depth and pH, are generally identified as the main
determinants of dominant plant species, species differentiation, and plant community composition in mires (Brancaleoni et al.,
2022; Laitinen et al., 2024). However, in our study, these factors could not explain the distribution of the vascular plant
communities. The distribution of bryophyte communities and the biomass of *Sphagnum* mosses, on the other hand, correlated
with these variables (Table B2). Indeed, the ratio of *Sphagnum* to brown mosses might be more significant in determining the
composition of vascular plant species in a fen, as most fen-specialist species prefer environments abundant in bryophytes other
than *Sphagnum* (Singh et al., 2019). This could explain the distribution of plant communities at our study site, as our analyses
showed that bryophytes had the strongest connection to the overall plant community composition when all plant species were
considered (Fig. B1). Additionally, there was an association between *Sphagnum* mosses and the fen generalist species *C.
rostrata* in the plant communities: the *C. rostrata*-cluster had the highest abundance of *Sphagnum* mosses among the three
vascular plant clusters (Table B1).

## 4.3 The role of plant biomass

The second hypothesis behind this study was that methane fluxes would be higher on plots where vascular plant biomass is
high, either in absolute terms or relative to bryophyte biomass. Our findings supported this, as both total vascular plant biomass
and the biomass ratio of vascular plants and bryophytes significantly correlated with methane fluxes during snow-free and
year-round periods (Fig. 6, Table B2). The strong correlation between total vascular plant biomass and the biomass of sedges
and *C. rostrata*, as well as the even stronger correlation of sedge and *C. rostrata* biomasses with methane fluxes (Table B2),
suggests, similar to earlier studies (e.g. Ge et al., 2023; Korrensalo et al., 2022), that plant functional type and species largely





determine the magnitude of the fluxes. High amounts of vascular plant, sedge, and *C. rostrata* biomass likely enhance methane production and release by supplying labile organic matter for methanogenesis and providing pathways for substrates to reach anoxic peat layers through deep root systems year-round (Alm et al., 1999; Joabsson et al., 1999, Saarinen, 1996). High flux rates from *C. rostrata* dominated plots (Fig. 7) may be due to the species' high transport rate (Ge et al., 2023), low oxidation potential (Ström et al., 2005), and the high porosity and large aerenchyma of its roots (Ge et al., 2023). High biomass of *C.*

*rostrata* may also drive winter methane fluxes, as this species is both biennial (Saarinen, 1998) and deep rooting, and can therefore provide labile carbon to deeper peat layers (Saarinen, 1996) supporting methane production year-round. Moreover, the significant correlation between the variable of BM ratio and methane fluxes brings a new perspective to the discussion, as previous studies have mainly focused on finding differences between single plant species (e.g. Bhullar et al., 2013; Ge et al., 2023; Koelbener et al., 2010; Korrensalo et al., 2022) or their role in a plant community (e.g. Lai et al., 2014; Riutta et al.,

2007; Ström et al., 2015). BM ratio could act as a parameter for predicting methane flux of peatlands and be used in remote sensing and modelling studies.

A majority (75 %) of the highest methane fluxes observed in this study originated from *C. rostrata* dominated plots. However, the magnitude of the fluxes was not solely dictated by the amount of *C. rostrata* biomass; the fluxes increased rapidly in a non-linear manner and only slightly with higher biomass, being relatively high and stable in most *C. rostrata-*

cluster plots (Fig. 7). As a species with high transport efficiency, a single *C. rostrata* shoot can transport a substantial volume of gases – potentially the same amount as a community with multiple shoots, where gas transport is distributed among many individuals (Koelbener et al., 2010; Korrensalo et al., 2022). This division of gas transport among more shoots could explain the saturation in the methane flux rates with the higher biomass of *C. rostrata*. Nevertheless, this does not imply that the size of *C. rostrata* shoots has no impact on the plant's transport efficiency. For the analyses, a mean biomass was used for each

species, and all individual shoots were assigned the same estimate weight. Consequently, an increase in biomass corresponds to a higher number of shoots.

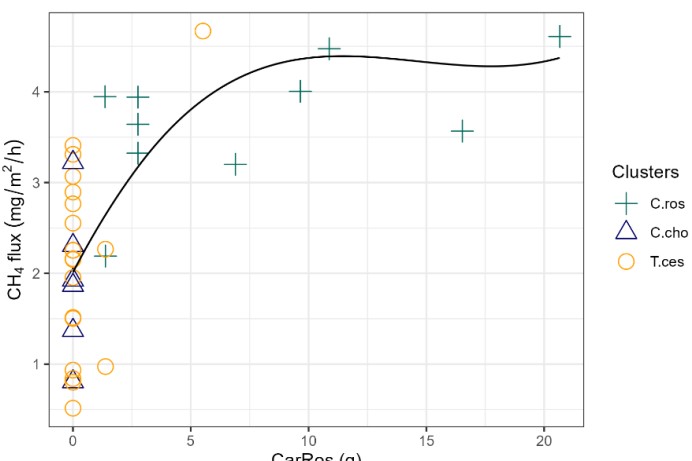

**Figure 7. The relation of snow-free season methane fluxes (mg/m²/h) and the biomass of *C. rostrata* (g of dry weight) (p < 0.001, r = 0.60) segregated by vascular plant clusters. See Fig. 5 for cluster abbreviations.**





### 4.4 Environmental factors explaining spatial variation of methane fluxes

We also asked if the spatial variability of methane fluxes could be explained by the combination of plant communities and environmental variables. Our results indicate that vegetation composition was the primary driver of this variability, as the significant relationship between methane fluxes and pH in snow free and year-round periods is likely also explained through vegetation (Jimenez-Alfaro et al., 2023; Keith et al., 2022). The positive correlation of $NO_3^-$ and $NO_2^-$ levels and methane fluxes during snow free and year-round periods was unexpected, as these compounds have usually been found to inhibit methane production (e.g. Knorr & Blodau, 2009). However, recent studies have debated their impact on methane production (Zhang et al., 2021). Water table depth, which is commonly thought to regulate methane fluxes in peatlands (Lai, 2009; Turetsky et al., 2014), did not correlate with methane fluxes at our site in any period. Indeed, in constantly wet fens with a stable water table, such as our study site, the depth of the water table may not control the variability of the fluxes (e.g. Ge et al., 2023). The thickness of the peat layer did not correlate with methane fluxes, which supports previous findings that most of methane released from peatlands is produced from the fresh root litter and root exudates instead of old, recalcitrant peat (Ström et al., 2012). However, peat depth had a significant correlation with the biomass of vascular plants, sedges, and *C. rostrata*, as well as the biomass ratio of vascular plants and bryophytes (Table B2), all of which were proxies for higher methane fluxes. Further, peat temperature did not correlate with the fluxes. All these findings highlight that vegetation, rather than environmental factors, was the main driver of methane fluxes at our site.

### Conclusions

Our manually measured year-round methane flux data from a northern boreal rich fen showed significant spatial and temporal variations in flux magnitude. Plant community composition, particularly the biomass of *Carex rostrata*, demonstrated a strong potential causality in explaining the plot-scale spatial variation of methane fluxes during snow free and year-round periods. Multivariate analysis also revealed a correlation between snow season fluxes and a vascular plant cluster most strongly associated with the presence of *C. rostrata*. These findings answer our first research question and support our hypothesis that plant community composition affects the flux. Environmental variables that significantly correlated with methane fluxes were pH and the combined concentrations of nitrate and nitrite in peat pore water during snow-free and year-round periods. These findings answer our second research question and indicate that plant community composition drives methane flux variability, both alone and in combination with other environmental factors. The total biomass of vascular plants and the ratio of vascular plant to bryophyte biomass also showed a significant positive relationship with methane fluxes in both year-round and snow-free seasons, confirming our second hypothesis that higher fluxes occur in plots with higher vascular plant biomass or a higher vascular plant to bryophyte biomass ratio. In addition to further research on plant properties, particularly species-specific traits (Ge et al., 2023), which is crucially needed for northern boreal fens, our findings suggest that the biomass ratio of vascular plants and bryophytes could potentially be used as a parameter for predicting methane emissions. This biomass ratio could also be valuable in remote sensing and modelling studies of peatlands with vegetation structure similar to our study site.



Importantly, these findings provide valuable insights for predicting realistic future changes in peatland methane emissions throughout the year, which is essential for estimating the potential impacts of ongoing climate change (Riutta et al., 2020).

## Appendices

Appendix A with additional site information. Appendix B with additional figures and tables of the results.

## Data availability

Full datasets used are available on request.

## Author contribution

MV, RP, MM and TRC designed the experimental setup of the study site. EJL, TT, TRC and JL designed the study. EJL
sampled the vegetation and performed the statistical analyses with assistance from JL, EK and TT. EJL, JL and EK were responsible for data curation. EK and MM wrote the code for the flux calculations. MV designed and provided resources for sampling the water chemistry. Resource management of the data collection were overseen by RP. Manual methane flux measurements, pore water sampling and water level measurements were conducted by numerous individuals, including EJL and EK. EJL prepared the manuscript with contributions from all co-authors.

**Competing interests**

The authors declare that they have no conflict of interest.

## Acknowledgements

Maik Bischoff, Pyry Runko, Noémie Perrier-Malette, Valentin Krieger, Polina Saarinen and Johannes Väisänen, among others, are gratefully acknowledged for their help with the fieldwork. A special thanks goes to the Oulanka Research station staff for
their continuous efforts. We also acknowledge that the help of AI was used when creating the R codes. EJL was supported by Green-Digi-Basin project funded by Research Council of Finland (RCF, 347704) and DIWA-Digital Waters flagship PhD pilot program funded by Ministry of Education (RCF, 359228). ELB, MM and TRC consider this study a contribution to GreenFeedBack (Greenhouse gas fluxes and earth system feedbacks) funded by the European Union's HORIZON research and innovation program under grant agreement No 101056921. Other funds during the project came from Maa- ja vesitekniikan
tuki ry (to MV). The work is part of EcoClimate experimental platform at Oulanka Research Station.



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





# Appendix A

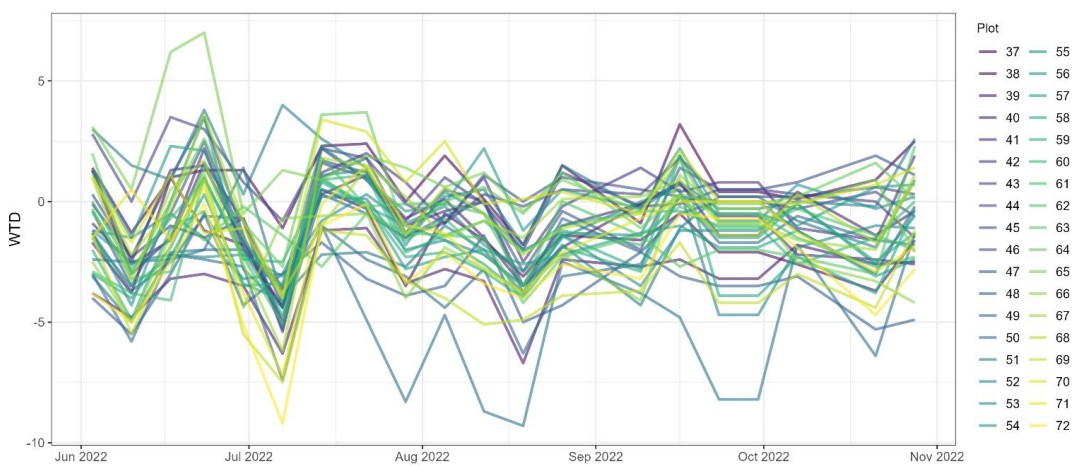

**Figure A1. The depth of the water table (WTD) at 36 study plots in Puukkosuo fen during June–October 2022. The average minimum value for WTD was -4.8 cm and the average maximum +1.8 cm from the peat surface.**

**Table A1. Distribution (Dist), mean (M), median (Mdn) and standard deviation (SD) of the heights (h) and dry biomass (BM) of vascular plant (VP) and bryophyte (Bryo) species identified from the 36 study plots (P) in July 2022. The amount of VPs within the studied area were counted by the number of shoots (Shoots_P). BMs are reported only for samples (S) collected for BM estimations. Separate samples were collected for fertile and sterile VP species, except for four species for which these were combined (Comb). The sample size (Sample) of bryophytes depended on the commonness of the species and it represented either 1 or 5 % of the studied**

**area. For all Bryo species a BM for 1 % coverage of the studied area was calculated (BM 1%). Abbreviations for the plant species:** *Andromeda polifolia* **(AndPol),** *Angelica sylvestris* **(AngSyl),** *Betula nana* **(BetNan),** *Carex chordorrhiza* **(CarCho),** *Carex dioica* **(CarDio),** *Carex flava* **(CarFla),** *Carex lasiocarpa* **(CarLas),** *Carex limosa* **(CarLim),** *Carex panicea* **(CarPan),** *Carex rostrata* **(CarRos),** *Dactylorhiza* **species (DactSp.),** *Drosera anglica* **(DroAng),** *Drosera rotundifolia* **(DroRot),** *Equisetum fluviatile* **(EquFlu),** *Equisetum variegatum* **(EquVar),** *Eriophorum angustifolium* **(EriAng),** *Eriophorum gracile* **(EriGra),** *Eriophorum latifolium* **(EriLat),**

*Festuca ovina* **(FesOvi),** *Menyanthes trifoliata* **(MenTri),** *Molinia caerulea* **(MolCae),** *Pedicularis palustris* **(PedPal),** *Pinguicula species* **(PingSp.),** *Potentilla erecta* **(PotEre),** *Saussurea alpina* **(SauAlp),** *Selaginella selaginoides* **(SelSel),** *Tofieldia pusilla* **(TofPus),** *Trichophorum alpinum* **(TriAlp),** *Trichophorum cespitosum* **(TriCes),** *Vaccinium oxycoccos* **(VacOxy),** *Viola epipsila* **(VioEpi),** *Aneura pinguis* **(AnePin),** *Aulacomnium palustre* **(AulPal),** *Campylium stellatum* **(CamSte),** *Cinclidium stygium* **(CinSty),** *Fissidens adianthoides* **(FisAdi),** *Mesoptychia rutheana* **(MesRut),** *Paludella squarrosa* **(PalSqu),** *Scorpidium cossonii* **(ScoCos),** *Sphagnum*

*warnstorfii* **(SphWar) and** *Tomentypnum nitens* **(TomNit). Plot-scale vegetation data can be obtained from the correspondent.**



| VP species | Fertility | Shoots_P (n) | h_Dist_P (cm) | h_Dist_S (cm) | h_M_P (cm) | h_M_S (cm) | h_Mdn_P (cm) | h_Mdn_S (cm) | SD_P (cm) | SD_S (cm) | BM_Dist (g) | BM_M (g) | BM_Mdn (g) | BM_SD (g) |
|---|---|---|---|---|---|---|---|---|---|---|---|---|---|---|
| AndPol | Comb | 217 | 3–9 | 3–8 | 5.4 | 4.8 | 5 | 4 | 1.6 | 2 | 0.0190–0.0941 | 0.0499 | 0.0394 | 0.03 |
| AngSyl | Sterile | 3 | 15 | 9–21 | 15 | 14.9 | 15 | 15.5 | 0 | 3.8 | 0.0356–0.2916 | 0.1317 | 0.1251 | 0.09 |
| BetNan | Comb | 48 | 4–16 | 4–17 | 8.8 | 10.4 | 9 | 10 | 3.6 | 3.5 | 0.0228–0.6131 | 0.1646 | 0.1192 | 0.17 |
| CarCho | Fertile | 122 | 13–22 | 11–22 | 15.6 | 16.4 | 15 | 16.5 | 2.8 | 3.2 | 0.0250–0.0824 | 0.0431 | 0.0359 | 0.02 |
| CarCho | Sterile | 596 | 12–22 | 11–19 | 14 | 14.7 | 13 | 14 | 2.6 | 2.5 | 0.0165–0.0611 | 0.0344 | 0.0295 | 0.01 |
| CarDio | Fertile | 6 | 9–28 | 12–24 | 17.3 | 17.8 | 15 | 16.5 | 9.7 | 3.5 | 0.0110–0.0359 | 0.0222 | 0.0227 | 0.01 |
| CarDio | Sterile | 141 | 8–15 | 9–16 | 9.8 | 12.3 | 9 | 12 | 2.1 | 2.6 | 0.0073–0.0326 | 0.0146 | 0.0126 | 0.01 |
| CarFla | Fertile | 2 | 18–35 | 28–44 | 26.5 | 32.8 | 26.5 | 32 | 12 | 5.1 | 0.1524–0.3685 | 0.2204 | 0.1908 | 0.06 |
| CarFla | Sterile | 7 | 18–23 | 16–29 | 20.5 | 23.8 | 20.5 | 25 | 3.5 | 4.7 | 0.0421–0.1581 | 0.1072 | 0.1045 | 0.04 |
| CarLas | Fertile | 1 | 53 | 27–62 | 53 | 52 | 53 | 55.5 | 0 | 11.8 | 0.1367–0.4719 | 0.3274 | 0.3528 | 0.1 |
| CarLas | Sterile | 457 | 37–63 | 41–66 | 48.7 | 55.4 | 48 | 54 | 6.3 | 8.7 | 0.1493–0.5235 | 0.2667 | 0.2483 | 0.11 |
| CarLim | Fertile | 3 | 18–20 | 20–28 | 19 | 24.4 | 19 | 24 | 1.4 | 3 | 0.0431–0.1291 | 0.0932 | 0.0986 | 0.03 |
| CarLim | Sterile | 31 | 15–18 | 15–24 | 16.5 | 19.8 | 16.5 | 20 | 2.1 | 3.4 | 0.0306–0.1366 | 0.0306 | 0.1366 | 0.03 |
| CarPan | Fertile | 3 | 23–24 | 13–42 | 23.5 | 29.8 | 23.5 | 30 | 0.7 | 11 | 0.0199–0.1986 | 0.1001 | 0.0943 | 0.05 |
| CarPan | Sterile | 45 | 15–36 | 14–38 | 23.1 | 25 | 23 | 26.5 | 6.4 | 8.1 | 0.0334–0.3339 | 0.1343 | 0.118 | 0.09 |
| CarRos | Fertile | 11 | 45–56 | 49–60 | 51 | 54.7 | 53 | 54 | 5.6 | 3.8 | 0.8980–1.8110 | 1.3596 | 1.3199 | 0.31 |
| CarRos | Sterile | 50 | 25–57 | 49–60 | 46 | 54.7 | 52 | 54 | 11 | 3.8 | 0.8980–1.8110 | 1.3788 | 1.3135 | 0.31 |
| DactSp. | Fertile | 1 | 24 | 24 | 24 | 24 | 24 | 24 | 0 | 0 | 0.4342 | 0.4342 | 0.4342 | 0 |
| DactSp. | Sterile | 8 | | | | | | | | | 0.0126–0.0486 | 0.0245 | 0.0222 | 0.01 |
| DroAng | Comb | 6 | | | | | | | | | 0.0025–0.0565 | 0.0191 | 0.0161 | 0.02 |
| DroRot | Comb | 18 | | | | | | | | | 0.0049–0.1010 | 0.0202 | 0.0099 | 0.03 |
| EquFlu | Sterile | 120 | 10–36 | 15–41 | 24.2 | 27.5 | 24 | 27 | 7.05 | 6.5 | 0.0580–0.2401 | 0.1387 | 0.1349 | 0.05 |
| EquVar | Sterile | 12 | 6–10 | 9–16 | 8 | 11.7 | 8 | 11.5 | 2.8 | 1.9 | 0.0258–0.0529 | 0.0355 | 0.0336 | 0.01 |
| EriAng | Sterile | 4 | 20–26 | 26–47 | 23.7 | 34.8 | 25 | 34 | 3.2 | 5.6 | 0.1752–0.4143 | 0.2841 | 0.2794 | 0.07 |
| EriGra | Fertile | 1 | 25 | 23–40 | 25 | 29.9 | 25 | 30.5 | 0 | 5 | 0.0509–0.1255 | 0.0761 | 0.0736 | 0.2 |
| EriGra | Sterile | 2 | 12 | 8–13 | 12 | 10.7 | 12 | 10.5 | 0 | 1.6 | 00305–0.0613 | 0.0437 | 0.0388 | 0.01 |
| EriLat | Sterile | 18 | 14–22 | 13–26 | 17.8 | 19.8 | 17.5 | 19.5 | 2.8 | 3.9 | 0.0523–0.3484 | 0.1556 | 0.1067 | 0.1 |
| FesOvi | Sterile | 2 | 7 | 7 | 7 | 7 | 7 | 7 | 0 | 0 | 0.0287–0.0307 | 0.0298 | 0.0298 | 0.001 |
| MenTri | Fertile | 2 | 7–8 | 16–23 | 7.5 | 18.1 | 7.5 | 19.5 | 0.7 | 2.1 | 0.1573–0.3780 | 0.2538 | 0.2322 | 0.1 |
| MenTri | Sterile | 180 | 3–10 | 5–14 | 6.5 | 9.4 | 6 | 9.5 | 2.1 | 3.3 | 0.0173–0.2177 | 0.0942 | 0.0628 | 0.07 |
| MolCae | Fertile | 1 | 33 | 31–74 | 33 | 55.7 | 33 | 56.5 | 0 | 3.3 | 0.1985–0.4128 | 0.3198 | 0.3343 | 0.07 |





| Species | Status | N | | | | | | | | | BM_Dist (g) | BM_M (g) | BM_Mdn (g) | BM_SD (g) |
|---|---|---|---|---|---|---|---|---|---|---|---|---|---|---|
| MolCae | Sterile | 173 | 21–36 | 21–36 | 25.5 | 28.4 | 24 | 29 | 4.7 | 5.9 | 0.0289–0.1339 | 0.0675 | 0.0527 | 0.04 |
| PedPal | Fertile | 5 | 12–21 | 10–37 | 15.6 | 23.1 | 14 | 24.5 | 3.8 | 8.2 | 0.1104–1.4398 | 0.6056 | 0.3976 | 0.49 |
| PedPal | Sterile | 7 | 4–14 | 5–14 | 9 | 7.8 | 9 | 6 | 7.1 | 3 | 0.0091–0.0616 | 0.0267 | 0.0153 | 0.02 |
| PingSp. | Sterile | 3 | | | | | | | | | 0.0074–0.0680 | 0.0312 | 0.0284 | 0.02 |
| PotEre | Fertile | 3 | 18 | 15–31 | 18 | 21 | 18 | 20 | 0 | 4.5 | 0.0836–0.9890 | 0.3105 | 0.2053 | 0.29 |
| PotEre | Sterile | 26 | 10–25 | 9616 | 14 | 12.5 | 11 | 12 | 5.7 | 2.5 | 0.0335–0.1021 | 0.0674 | 0.0722 | 0.02 |
| SauAlp | Sterile | 3 | 13 | 5–22 | 13 | 12.8 | 13 | 13 | 0 | 4.7 | 0.0141–0.3586 | 0.1127 | 0.0806 | 0.1 |
| SelSel | Fertile | 31 | 4–6 | 2–9 | 4.6 | 5.4 | 4 | 5 | 0.9 | 2.2 | 0.0074–0.0310 | 0.017 | 0.0154 | 0.01 |
| TofPus | Fertile | 1 | 21 | 13–24 | 21 | 17.9 | 21 | 17 | 0 | 3.6 | 0.0182–0.1026 | 0.0564 | 0.0558 | 0.03 |
| TofPus | Sterile | 30 | | | | | | | | | 0.0065–0.0316 | 0.0191 | 0.0213 | 0.01 |
| TriAlp | Fertile | 71 | 13–25 | 15–24 | 18.4 | 20.2 | 17.5 | 21.5 | 3.5 | 3.2 | 0.0128–0.0332 | 0.022 | 0.0223 | 0.01 |
| TriCes | Fertile | 1828 | 6–27 | 15–27 | 19.4 | 20.5 | 20 | 20 | 5.3 | 3.2 | 0.0168–0.0379 | 0.0247 | 0.0245 | 0.01 |
| VacOxy | Sterile | 86 | 3–15 | 3–13 | 6.8 | 7.8 | 5 | 6.5 | 3.6 | 4 | 0.0057–0.0326 | 0.0175 | 0.0177 | 0.01 |
| VioEpi | Sterile | 4 | 3–5 | 4–8 | 4 | 6.3 | 4 | 6.5 | 1.4 | 1.4 | 0.0065–0.0378 | 0.0228 | | 0.01 |

| Bryo species | Sample | BM_Dist (g) | BM_M (g) | BM_Mdn (g) | BM_SD (g) | BM 1% (g) |
|---|---|---|---|---|---|---|
| AnePin | 1 x 1 % | 0.0126 | 0.0126 | 0.0126 | 0 | 0.0126 |
| AulPal | 3 x 5 % | 0.6562–1.2110 | 0.9212 | 0.8965 | 0.28 | 0.1842 |
| CamSte | 3 x 5 % | 0.3650–0.4697 | 0.4172 | 0.417 | 0.05 | 0.0834 |
| CinSty | 1 x 1 % | 0.0307 | 0.0307 | 0.0307 | 0 | 0.0307 |
| FisAdi | 1 x 1 % | 0.0691 | 0.0691 | 0.0691 | 0 | 0.0691 |
| MesRut | 1 x 1 % | 0.0264 | 0.0264 | 0.0264 | 0 | 0.0264 |
| PalSqu | 1 x 1 % | 0.0695 | 0.0695 | 0.0695 | 0 | 0.0695 |
| ScoCos | 3 x 5 % | 0.5623–1.0105 | 0.777 | 0.7582 | 0.22 | 0.1554 |
| SphWar | 3 x 5 % | 0.5028–1.1476 | 0.6669 | 0.8417 | 0.32 | 0.1334 |
| TomNit | 3 x 5 % | 0.5402–0.9949 | 0.7166 | 0.6148 | 0.24 | 0.1433 |



**Table A2. Total plot-scale dry mass weight biomasses (BM) of vascular plants (VP) and bryophytes (Bryo) in grams inside the collar**
**(A=660.5 cm$^2$) at each experimental plot (37–72), a VP to Bryo BM ratio (VP/Bryo ratio) and the portion of VP BM of total BM (VP of total BM).**

| Plot | BM VP | BM Bryo | VP/Bryo ratio | VP of total BM |
|------|-------|---------|---------------|----------------|
| 37 | 17.8538 | 15.0746 | 118 % | 54 % |
| 38 | 22.0626 | 17.7820 | 124 % | 55 % |
| 39 | 6.2854 | 13.3400 | 47 % | 32 % |
| 40 | 7.9262 | 8.0765 | 98 % | 50 % |
| 41 | 6.4044 | 8.0765 | 79 % | 44 % |
| 42 | 16.8502 | 9.1706 | 184 % | 65 % |
| 43 | 10.6523 | 13.0900 | 81 % | 45 % |
| 44 | 9.7756 | 11.7626 | 83 % | 45 % |
| 45 | 8.9990 | 10.1400 | 89 % | 47 % |
| 46 | 10.4179 | 8.7790 | 119 % | 54 % |
| 47 | 14.3059 | 12.8400 | 111 % | 53 % |
| 48 | 7.5385 | 13.8365 | 54 % | 35 % |
| 49 | 6.6854 | 4.2710 | 157 % | 61 % |
| 50 | 32.1099 | 12.4200 | 259 % | 72 % |
| 51 | 15.2690 | 8.8400 | 173 % | 63 % |
| 52 | 11.7328 | 9.3945 | 125 % | 56 % |
| 53 | 9.5022 | 15.5400 | 61 % | 38 % |
| 54 | 9.1127 | 8.3400 | 109 % | 52 % |
| 55 | 9.9275 | 11.7626 | 84 % | 46 % |
| 56 | 8.2885 | 9.7800 | 85 % | 46 % |
| 57 | 10.3305 | 8.7000 | 119 % | 54 % |
| 58 | 4.1176 | 15.1273 | 27 % | 21 % |
| 59 | 5.3803 | 13.3400 | 40 % | 29 % |
| 60 | 6.7374 | 10.9706 | 61 % | 38 % |
| 61 | 8.9244 | 15.5400 | 57 % | 36 % |
| 62 | 8.2568 | 8.8896 | 93 % | 48 % |
| 63 | 11.0370 | 7.7520 | 142 % | 59 % |
| 64 | 10.2857 | 11.2200 | 92 % | 48 % |
| 65 | 13.5974 | 11.8010 | 115 % | 54 % |
| 66 | 8.2232 | 15.1800 | 54 % | 35 % |
| 67 | 5.9341 | 15.5400 | 38 % | 28 % |
| 68 | 7.2415 | 15.2906 | 47 % | 32 % |
| 69 | 8.8783 | 13.3400 | 67 % | 40 % |
| 70 | 11.2556 | 11.6765 | 96 % | 49 % |
| 71 | 7.4490 | 12.2400 | 61 % | 38 % |
| 72 | 9.0483 | 13.3400 | 68 % | 40 % |



**Table A3. Combined dry mass weight estimation (g) of all the study plots for each species. Vascular plants on the top, bryophytes at the bottom of the list in alphabetical order. For abbreviations of the species, see Table A1.**

| Species | Mass (g) |
|---|---|
| CarLas | 121.9426 |
| CarRos | 83.8956 |
| TriCes | 40.2116 |
| CarCho | 25.7606 |
| MenTri | 17.5578 |
| MolCae | 16.6548 |
| EquFlu | 16.644 |
| AndPol | 11.3273 |
| BetNan | 7.9008 |
| TriAlp | 6.502 |
| CarPan | 6.3438 |
| PedPal | 4.4261 |
| PotEre | 3.2905 |
| EriLat | 3.2676 |
| CarDio | 2.1918 |
| VacOxy | 1.505 |
| CarLim | 1.2282 |
| CarFla | 1.1912 |
| EriAng | 1.1364 |
| TofPus | 1.0885 |
| SauAlp | 1.0143 |
| SelSel | 0.663 |
| DactSp. | 0.6302 |
| EquVar | 0.426 |
| AngSyl | 0.3951 |
| DroRot | 0.3636 |
| PingSp. | 0.1872 |
| EriGra | 0.1635 |
| DroLon | 0.1146 |
| VioEpi | 0.0912 |
| FesOvi | 0.0596 |
| ScoCos | 192.8514 |
| CamSte | 112.5066 |
| SphWar | 91.6458 |
| AulPal | 18.2358 |
| CinSty | 2.0262 |
| MesRut | 1.584 |
| PalSqu | 1.529 |
| FisAdi | 1.1056 |
| TomNit | 0.7166 |
| AnePin | 0.063 |



# Appendix B

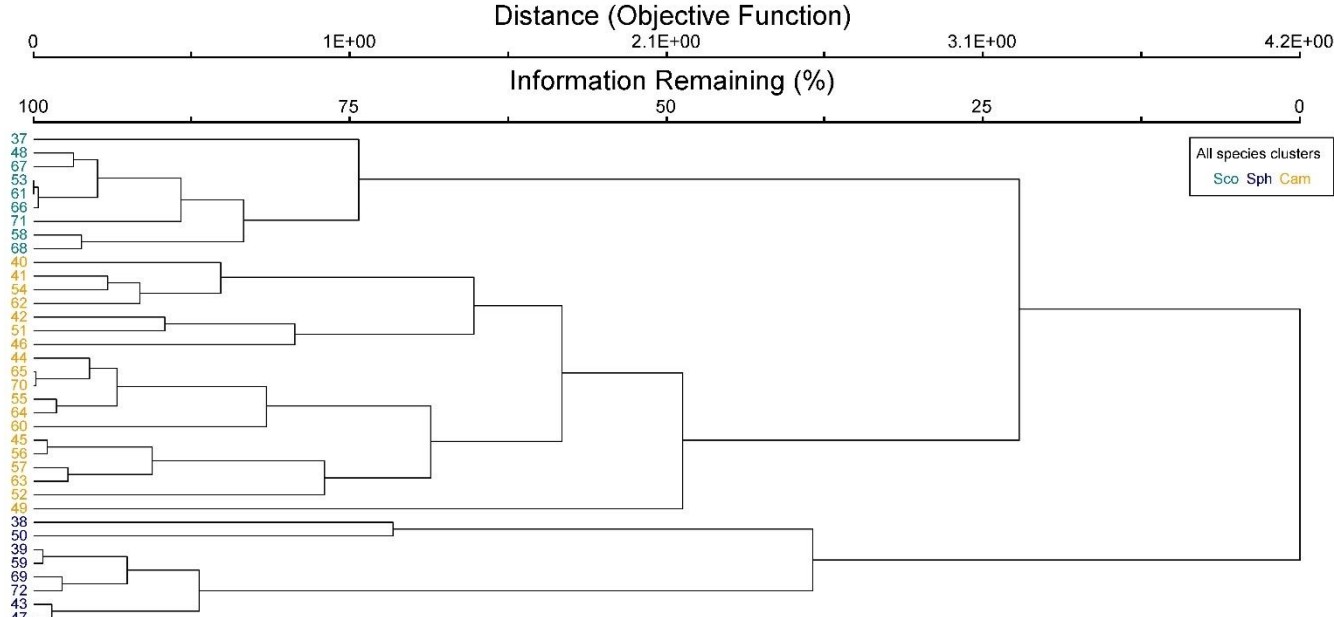

**Figure B1. Cluster dendrogram of all species-data. Indicator species by clusters: Sco = *Scorpidium cossonii*, Sph = *Sphagnum* spp. and *Paludella squarrosa* and Cam = *Campylium stellatum*.**



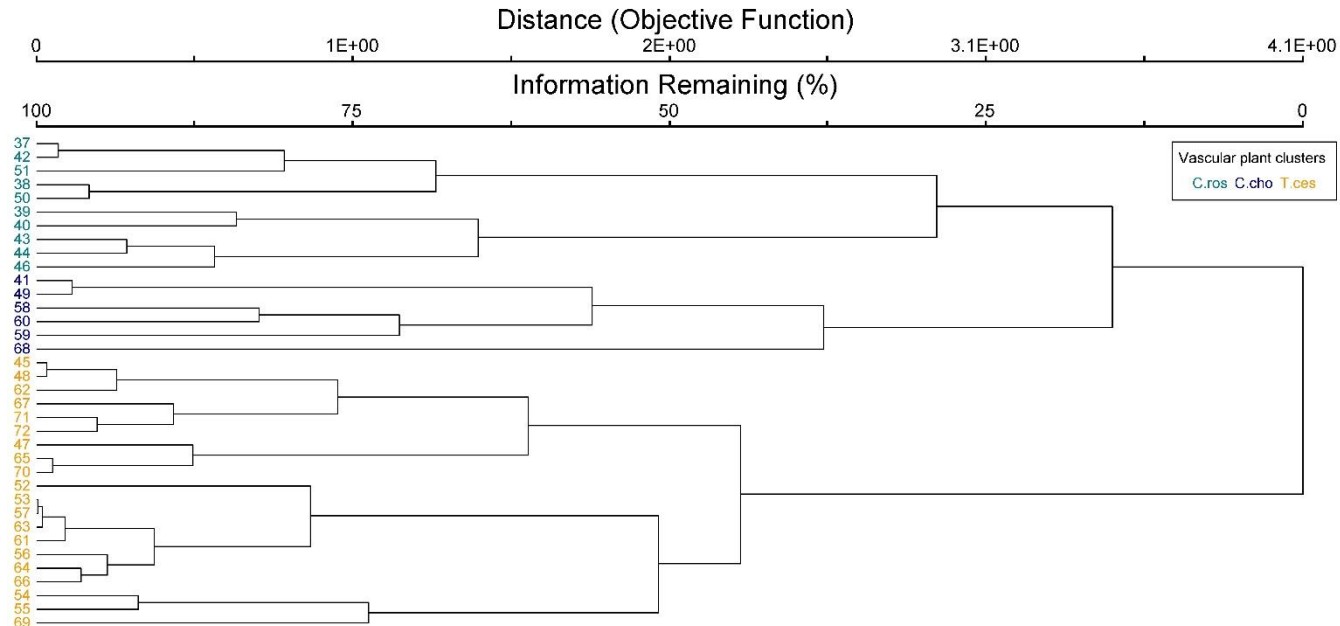

**Figure B2. Cluster dendrogram of vascular plant data. Indicator species by clusters: C.ros=** *Carex rostrata,* **C.cho=** *Carex chordorrhiza* **and T.ces =** *Carex lasiocarpa, Potentilla erecta* **and** *Trichophorum cespitosum.*

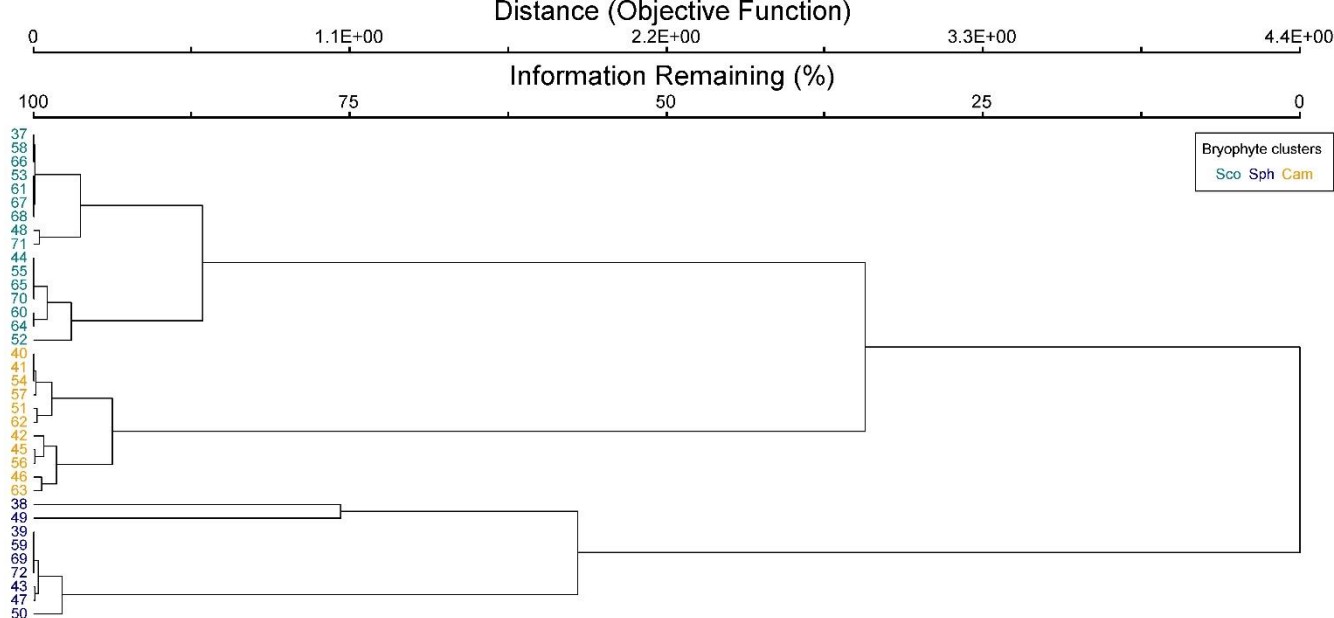

**Figure B3. Cluster dendrogram of bryophyte data. Indicator species by clusters: Sco =** *Scorpidium cossonii*, **Sph =** *Sphagnum* **spp. and** *Paludella squarrosa* **and Cam =** *Campylium stellatum.*





**Table B1. Indicator values (% of perfect indication, based on combining the values of relative abundance and relative frequency) of all identified species (n = 40) of Puukkosuo segregated by the vascular plant clusters. The values for the clusters' significant indicator species are bolded. Species with indicator value lower than 20 % were excluded from the table. Abbreviations for the species most strongly connected to the clusters: C.ros = *C. rostrata*, C.cho = *C. chordorrhiza* and T. ces = *C. lasiocarpa*, *P. erecta*, and *T. cespitosum*. Abbreviations for the species: CarRos = *Carex rostrata*, MenTri = *Menyanthes trifoliata*, TriAlp =*Trichophorum alpinum*, CarLim = *Carex limosa*, CinSty = *Cinclidium stygiym*, CarCho = *Carex chordorrhiza*, VacOxy = *Vaccinium oxycoccos*, BetNan = *Betula nana*, AndPol = *Andromeda polifolia*, EriLat = *Eriphorum latifolium*, TriCes = *Trichophorum cespitosum*, PotEre = *Potentilla erecta*, CarLas = *Carex lasiocarpa*, EquFlu = *Equisetum fluviatile*, MolCae = *Molinia caerulea*, TofPus = *Tofieldia pusilla*, PingSp. = *Pinguicula* sp., CarDio = *Carex dioica*, CamSte = *Campylium stellatum*, ScoCos = *Scorpidium cossonii* and SphWar = *Sphagnum warnstorfii*.**

| | C.ros | C.cho | T.ces |
|---|---|---|---|
| **C.ros** | | | |
| CarRos (p < 0.01) | **95** | 0 | 1 |
| MenTri | 38 | 16 | 18 |
| TriAlp | 31 | 2 | 8 |
| CarLim | 20 | 0 | 0 |
| **C.cho** | | | |
| CinSty (p = 0.01) | 8 | **64** | 2 |
| CarCho (p = 0.02) | 31 | **58** | 5 |
| VacOxy | 9 | 46 | 11 |
| BetNan | 26 | 39 | 3 |
| AndPol | 31 | 36 | 25 |
| EriLat | 7 | 31 | 2 |
| **T.ces** | | | |
| TriCes (p < 0.01) | 7 | 13 | **65** |
| PotEre (p < 0.01) | 0 | 0 | **55** |
| CarLas (p < 0.01) | 36 | 11 | **51** |
| EquFlu | 35 | 8 | 41 |
| MolCae | 6 | 15 | 41 |
| TofPus | 4 | 1 | 27 |
| PingSp. | 0 | 0 | 25 |
| CarDio | 0 | 18 | 23 |
| **Relative frequencies of common species** | | | |
| CamSte | 70 | 67 | 75 |
| VacOxy | 40 | 83 | 45 |
| ScoCos. | 40 | 50 | 75 |
| SphWar | 60 | 17 | 20 |





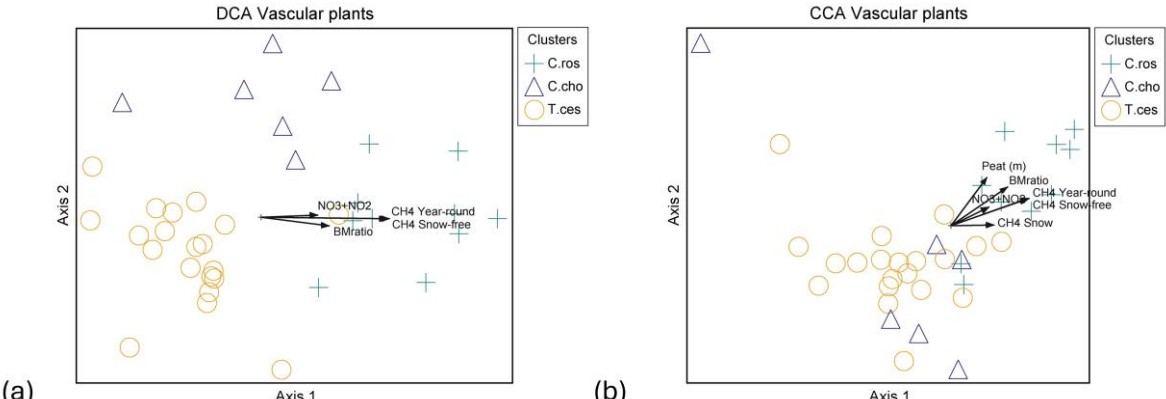

**Figure B4. Ordination graphs of vascular plant data. (a) DCA with rare species downweighed (eigenvalue axis 1 = 0.463, axis 2 = 0.154) and (b) CCA (eigenvalue axis 1 = 0.423, axis 2 = 0.163). Plant community clusters: C.ros = *Carex rostrata*, C.cho = *Carex chordorrhiza* and T.ces = *Carex lasiocarpa*, *Potentilla erecta* and *Trichophorum cespitosum*. Timespans for methane fluxes: CH4 Year-round (19.10.2021–31.10.2022), CH4 Snow-free (13.5.–26.10.2022) and CH4 Snow (19.10.2021–12.5.2022, 27.-31.10.2022). NO3+NO2 is the combined concentration of nitrate (NO₃⁻) and nitrite (NO₂⁻), BMratio is the ratio of vascular plant and bryophyte biomasses and Peat (m) is the peat layer thickness.**

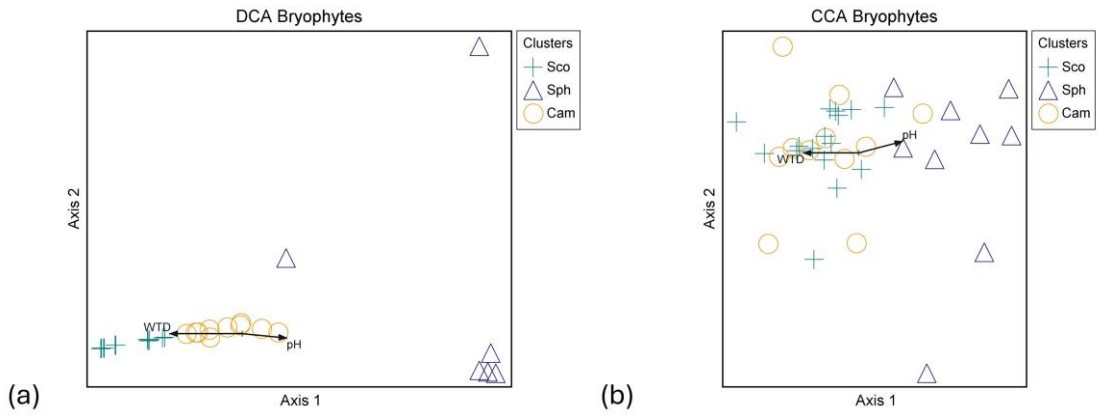

**Figure B5. Ordination graphs of bryophyte data. (a) DCA with rare species downweighed (eigenvalue axis 1 = 0.950, axis 2 = 0.364) and (b) CCA (eigenvalue axis 1 = 0.711, axis 2 = 0.510). Abbreviations for plant community clusters: Sco = *Scorpidium cossonii*, Sph = *Sphagnum* spp. and *Paludella squarrosa* and Cam = *Campylium stellatum*. WTD stands for water table depth and CarRos for biomass of *C. rostrata*.**





**Table B2.** Correlation matrix among the environmental and biomass variables. Significant correlations bolded, p-value ≤ 0.05 (r ≥ 0.33) marked with one asterisk (*), p-value ≤ 0.01 (r ≥ 0.42) with two asterisks (**) and p-value ≤ 0.001 (r ≥ 0.53) with three asterisks (***). CH4_y = year-round methane flux, CH4_sf = snow-free season methane flux, CH4_s = snow season methane flux, WTD = snow free season water table depth, snow free season concentrations of NO3+NO2 = nitrate and nitrite, NH4 = ammonium, TOTN = total nitrogen, TOC = total organic carbon, IC = inorganic carbon, BMratio = vascular plant to bryophyte biomass ratio, BMvp = vascular plant biomass, BMbryo = bryophyte biomass, BMsed = biomass of all sedges, BMsph = biomass of *Sphagnum* mosses, Peat = peat layer thickness, Litter = litter percentage, CarRos = biomass of *Carex rostrata*, SoilT_y = average year-round soil temperature at -5 cm, SoilT_sf = average snow-free season soil temperature at -5 cm and SoilT_s = average snow season soil temperature at -5 cm.

| | CH4_y | CH4_sf | CH4_s | WTD | pH | NO3+NO2 | NH4 | TOTN | TOC | IC | BMratio | BMvp | BMbryo | BMsed | BMsph | Peat | Litter | C.Ros | SoilT_y | SoilT_sf | SoilT_s |
|---|---|---|---|---|---|---|---|---|---|---|---|---|---|---|---|---|---|---|---|---|---|
| CH4_y | | | | | | | | | | | | | | | | | | | | | |
| CH4_sf | 0.99*** | | | | | | | | | | | | | | | | | | | | |
| CH4_s | 0.60*** | 0.55*** | | | | | | | | | | | | | | | | | | | |
| WTD | -0.05 | -0.06 | | | | | | | | | | | | | | | | | | | |
| pH | -0.41* | -0.38* | | -0.48** | | | | | | | | | | | | | | | | | |
| NO3+NO2 | 0.34* | 0.34* | 0.15 | 0.15 | -0.11 | | | | | | | | | | | | | | | | |
| NH4 | 0.17 | 0.17 | | -0.18 | 0.23 | 0.20 | | | | | | | | | | | | | | | |
| TOTN | 0.21 | 0.22 | 0.10 | -0.04 | -0.21 | -0.18 | -0.04 | | | | | | | | | | | | | | |
| TOC | 0.24 | 0.22 | 0.07 | -0.07 | -0.28 | 0.05 | -0.07 | 0.08 | | | | | | | | | | | | | |
| IC | -0.08 | -0.08 | 0.13 | 0.03 | -0.05 | -0.14 | 0.03 | -0.14 | 0.41* | | | | | | | | | | | | |
| BMratio | 0.45** | 0.45** | 0.24 | -0.13 | -0.30 | 0.13 | 0.03 | 0.36* | 0.28 | 0.09 | | | | | | | | | | | |
| BMvp | 0.53*** | 0.54*** | 0.16 | -0.17 | -0.26 | 0.14 | -0.10 | 0.23 | 0.22 | 0.20 | 0.81*** | | | | | | | | | | |
| BMbryo | 0.03 | 0.05 | -0.19 | -0.13 | 0.16 | -0.07 | 0.13 | -0.16 | -0.10 | 0.17 | -0.46** | 0.11 | | | | | | | | | |
| Bmsedg | 0.57*** | 0.57*** | 0.21 | -0.15 | -0.35 | 0.14 | -0.09 | 0.24 | 0.19 | 0.15 | 0.78*** | 0.98*** | 0.14 | | | | | | | | |
| BMsph | 0.16 | 0.19 | 0.11 | -0.59*** | 0.55*** | -0.09 | 0.13 | -0.09 | -0.03 | -0.17 | -0.02 | 0.13 | 0.22 | 0.06 | | | | | | | |
| Peat | 0.09 | 0.07 | 0.17 | -0.20 | 0.12 | 0.28 | 0.15 | 0.12 | 0.34* | 0.31 | 0.37* | 0.37* | -0.03 | 0.34* | 0.12 | | | | | | |
| Litter | 0.12 | 0.12 | 0.09 | 0.00 | -0.15 | -0.03 | 0.34* | 0.09 | 0.20 | 0.11 | 0.46** | 0.59*** | 0.06 | 0.62*** | -0.02 | 0.22 | | | | | |
| C.Ros | 0.61*** | 0.60*** | 0.20 | -0.23 | -0.22 | 0.24 | 0.01 | 0.29 | 0.13 | 0.11 | 0.71*** | 0.93*** | 0.19 | 0.93*** | 0.16 | 0.40* | 0.48** | | | | |
| SoilT_y | 0.22 | 0.17 | 0.23 | 0.19 | -0.25 | 0.07 | -0.01 | 0.06 | -0.01 | -0.13 | -0.06 | 0.03 | 0.03 | 0.01 | -0.12 | -0.01 | -0.25 | 0.11 | | | |
| SoilT_sf | 0.06 | 0.06 | 0.24 | 0.21 | -0.07 | 0.03 | -0.15 | 0.10 | -0.10 | -0.21 | -0.14 | -0.08 | 0.06 | -0.12 | 0.02 | -0.06 | -0.23 | -0.08 | 0.62*** | | |
| SoilT_s | -0.29 | -0.27 | -0.14 | -0.21 | 0.36 | -0.12 | 0.28 | -0.19 | -0.09 | 0.07 | -0.15 | -0.09 | 0.16 | -0.14 | 0.09 | 0.09 | -0.22 | -0.13 | 0.03 | 0.09 | |

40



**Table B3. Accumulated annual (AccAnnCH4) and snow season (AccSnowCH4) methane flux rates and the portion of snow season**
45  **fluxes of the annual flux (%SnowOfAnnual) calculated for each plot from the manual static chamber measurements.**

| Plot | AccAnnCH4 | AccSnowCH4 | %SnowOfAnnual |
|------|-----------|------------|---------------|
| 41 | 6859.75 | 1462.88 | 21.33 |
| 64 | 10490.26 | 2050.96 | 19.55 |
| 71 | 3907.02 | 724.26 | 18.54 |
| 48 | 14210.27 | 2261.19 | 15.91 |
| 72 | 3901.88 | 605.24 | 15.51 |
| 46 | 15486.28 | 2307.50 | 14.90 |
| 49 | 8821.98 | 1256.22 | 14.24 |
| 67 | 2262.12 | 244.18 | 10.79 |
| 55 | 11231.20 | 1202.56 | 10.71 |
| 69 | 3623.34 | 375.92 | 10.37 |
| 62 | 6730.06 | 660.54 | 9.81 |
| 58 | 14024.16 | 1372.14 | 9.78 |
| 68 | 3992.76 | 342.10 | 8.57 |
| 52 | 12463.48 | 981.63 | 7.88 |
| 59 | 9800.62 | 771.15 | 7.87 |
| 57 | 9525.56 | 745.81 | 7.83 |
| 40 | 9417.67 | 725.66 | 7.71 |
| 70 | 4104.88 | 313.75 | 7.64 |
| 42 | 19409.26 | 1451.68 | 7.48 |
| 63 | 9436.37 | 678.56 | 7.19 |
| 38 | 15134.87 | 1037.47 | 6.85 |
| 39 | 16616.16 | 1113.47 | 6.70 |
| 56 | 14225.52 | 939.73 | 6.61 |
| 50 | 19267.61 | 1263.10 | 6.56 |
| 43 | 16525.27 | 1078.93 | 6.53 |
| 65 | 9711.21 | 620.80 | 6.39 |
| 54 | 6300.25 | 387.94 | 6.16 |
| 51 | 13298.34 | 750.18 | 5.64 |
| 47 | 19544.40 | 1064.65 | 5.45 |
| 61 | 9024.76 | 486.75 | 5.39 |
| 60 | 7824.93 | 414.35 | 5.30 |
| 37 | 16767.20 | 823.34 | 4.91 |
| 44 | 15123.26 | 640.19 | 4.23 |
| 53 | 8097.61 | 339.17 | 4.19 |
| 45 | 13906.22 | 419.71 | 3.02 |
| 66 | 11038.68 | 250.06 | 2.27 |
| TOTAL | 392105.20 | 32163.79 | 8.20 |





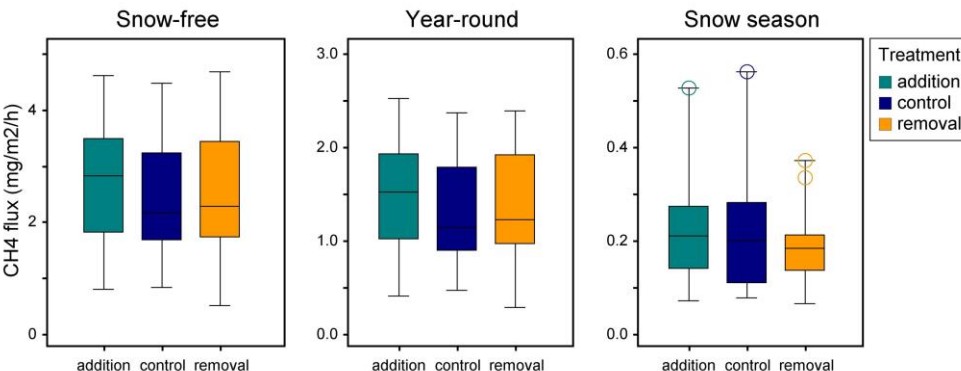

**Figure B6. The relation of snow level manipulation experiment's treatments and methane flux in different time periods. There were no significant (p < 0.05) differences between the treatments in relation to the fluxes. Control plots represented the natural snow level, while in removal plots, the snow depth was maintained at 0.25 m by shovelling the snow and distributing it evenly onto the addition plots.**

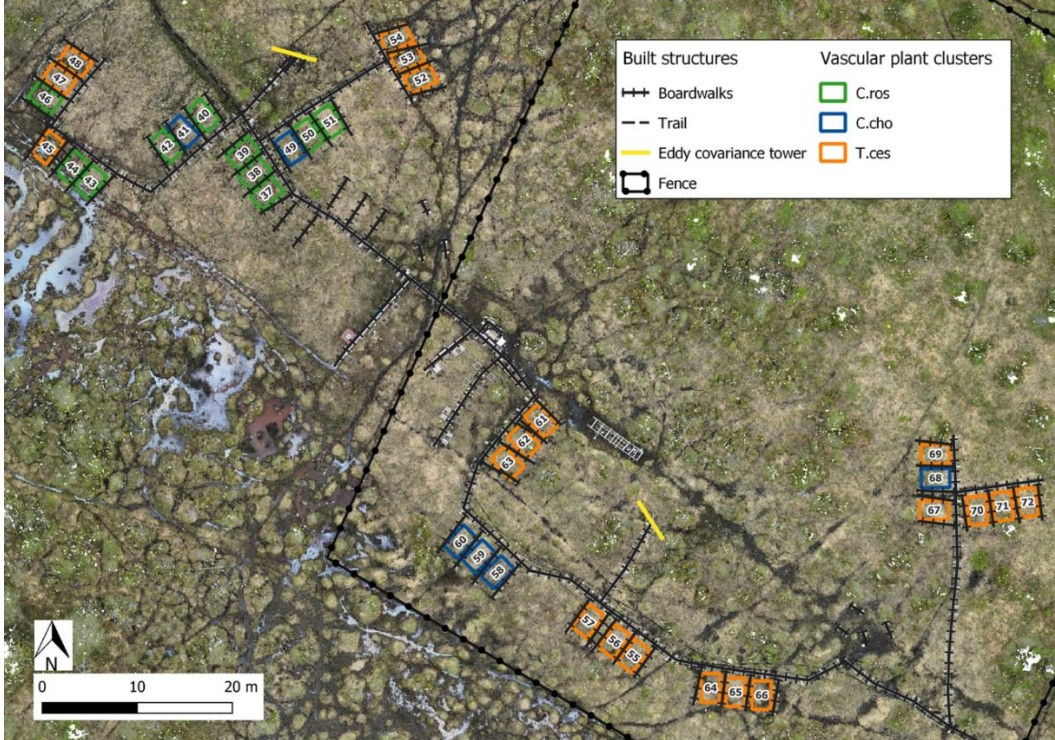

**Figure B7. Location of the study plots at Puukkosuo fen segragated by vascular plant clusters. Indicator species most strongly connected to the vascular plant clusters: C.ros = *C. rostrata*, C.cho = *C. chordorrhiza* and T. ces = *C. lasiocarpa*, *P. erecta*, and *T. cespitosum*. Orthomosaic © Petra Korhonen 2024.**