# Peer review of "Plant community composition controls spatial variation in year-round methane fluxes in a boreal rich fen"

_EGUsphere, 2025_

## Author Response (AR1)

**REVIEW COMMENTS #1**

**General comments:**

This manuscript by Järvi-Laturi et al, looks into fine-scale spatial variation in methane flux in a boreal peatland over a full year. By measuring species-specific vascular plant and moss biomass together with chamber-based methane flux measurements, the authors found that increasing sedge biomass, in particular that of *Carex rostrata*, seem to increase methane emissions significantly especially in the snow-free season and full-year scales. Based on their analyses, vegetation composition and biomass seemed to be a stronger driver of methane fluxes than abiotic environmental variables at this site. The authors attributed these results to both enhanced provision of carbon substrates for methanogens and methane transport from soil to the atmosphere.

This is a study that provides the much-needed data and overview of both wintertime and full-year methane fluxes in a peatland, data that still to this day are quite scarce and thus valuable. As the authors mention in the manuscript, regional and global wetland methane budgets contain large uncertainties, some of which are related to the spatial variation in methane fluxes and vegetation composition. Therefore, this study, which looks at small scale (between-plot) methane flux variation, has potential in adding to our understanding of plant-mediated methane emissions in carbon-rich peatland ecosystems. While I see a lot of value and potential in this work, I recommend a list of improvements (major or minor, depending on how biomass measurements were conducted):

Response: We thank you for the valuable feedback and corrections and appreciate the positive comments. Below, we have addressed the comments.

**Specific comments:**

- 1. My main criticism is related to the plant biomass measurements and their representativeness of the chamber collars:
  - a) It is unclear how you scaled the vascular plant biomass measurements to the collars. You mention that you counted the number of shoots per species within the collar, but did you use this shoot number to scale the mean biomass of 10 samples to the actual collar (or did you actually take average of 20 sample plants if you ignored the fertile/sterile division? See comment 1 c)? If you did not scale these measurements to the collar, you cannot reliably estimate the collar species biomass, and so I recommend to do this and re-analyze your data and fix the results in 3.2, 3.3, 3.4, 3.5 with appropriately scaled biomass values. If you did do this, please explain this clearly and in more detail in the methods section.

Response: We agree that biomass sampling and scaling were not clearly enough reported. There has been a misunderstanding of the methodology which we have now clarified in the manuscript. Below, we explain the process.

Firstly, we counted shoots per vascular plant species inside each collar and also measured mean height of each species. After this we estimated the percentage coverage of bryophytes. Please, see Table A1 which shows the total number/cover of vascular plants/bryophytes inside all collars and their statistical parameters.

Secondly, we collected biomass samples for each species from outside the research area to avoid disturbance to the experimental area. These areas located within approximately 50 m range north from plots 46-48 and 52-54 (Fib. B7).

For vascular plants, the biomass sampling was randomized so that we first selected an area where vegetation heights resembled the heights of the vegetation within the collars. Then, we randomly tossed a marker and selected the first ten non-flowering individuals of target vascular plant species close to the marker. We additionally collected ten flowering individuals for those species, which were found flowering inside the collars. With uncommon species, randomization could not be put into practice (Angelica sylvestris, Carex dioica, Carex panicea, Dactylorhiza sp., Drosera sp., Eriophorum angustifolium, Festuca ovina, Pinguicula sp., Saussurea alpina and Viola epipsila). For these, samples were collected from where they could be found. We believe that our vascular plant biomass samples represent well the vascular vegetation within the collars (Table A1).

For bryophytes, the biomass samples were collected around the experimental area, within approximately 50 m distance from the plots. Sample size was determined by the bryophyte species, being either 5 % or 1 % of the collar area (see species Table A1). The diameter of the 5 % sample was 6.6 cm (three replicates), and of the 1 % sample 2.95 cm (one sample). The sampling locations were selected so that the target bryophyte species could be found as "pure monoculture" as possible.

As we have explained also in the manuscript, the biomass samples were dried and weighed and the dry weights were normalized either by shoot (vascular plant, g/shoot) or by cover percent (bryophyte, g/1%). Then for each vascular plant species we used the total number of shoots within a collar multiplied by the mean dry mass per shoot to obtain total species biomass for each collar. For each bryophyte species, we used the total percentage coverage within a collar multiplied by the mean dry mass per 1% area to obtain total species biomass for each collar.

We acknowledge that we do not gain exact biomass values / species / collar with this method. However, we believe this method captures the fine-scale variation between the individuals as both lower and higher plant individuals were sampled for biomass determination and the height variation resembled the variation within the collars.

b) It is also unclear how you took the moss biomass samples and scaled them to the collar. How did you determine which species were "most common" and which were not? What was the percentage cover limit (if there was one)? What was the spatial scale that you used for estimating the "most common" species- was it across the whole study site or within individual collar? If it was across the whole study site, I don't quite understand the logic of taking a biomass sample equaling to 5 % of the collar area (i.e. 33.025 cm2) especially if the collar had an actual percentage cover <5%, in which case you may have overestimated the species biomass in the collar. Or did all of these "most common" species have >5% coverage in all collars?

If you looked at this at the scale of individual collars, did you take a sample that was equal to 5 % of the collar area per species for the five most common species within each collar and for the rest of the species found within the collar, you took a sample over an area equal to 1% area of the collar (i.e. 6.605 cm2)? What did you do if there were less than five species within the collar? Did every collar really have 10 species within them (now the text kind of makes it sound like there were but it doesn't seem likely to me)? Please specify this in the methods.

And, most importantly, how did you scale the moss biomass samples to the percentage coverage within the individual collars? As with the vascular plant biomass, if no scaling was done, the moss biomass measurements do not represent the actual collar moss biomass and the data should be reanalyzed with appropriate scaling. Please specify this clearly in the methods.

Response: Our aim was not to determine the bryophyte species by their commonness, and therefore we have removed the mention of this classification from the revised manuscript. Above, we have explained the process of bryophyte biomass sampling in more detail.

c) How did you determine the locations for vascular plant and moss biomass sampling? Were the soil conditions (e.g. pH, soil moisture) similar to the collar? Did you look at and compare the general species composition in the collar vs the plots where you collected the biomass samples (between-species competition could affect some of the plant trait expression and thus biomass), for example by determining percentage cover? How did you decide which plants and moss patches to pick?

For mosses, did you look at e.g. moss stem density in some way to try to estimate the moss biomass in the collar and in the sampling points more reliably than just percentage coverage (the same percentage coverage can represent very different moss biomasses in different collars due to variation in moss stem density and other structural properties)? If not, the moss biomass estimates may be very uncertain and I would recommend discussing these uncertainties explicitly and in much more detail in the manuscript. Given these uncertainties, I would also recommend not to emphasize the ratio between vascular plants and bryophytes as an important methane flux predictor as much as you have so far in this manuscript, or at least combine it with adequate discussion about its uncertainties.

If you did not estimate the similarity (in terms of abiotic/biotic variables) between the collar and the plot where you collected the representative biomass samples, I would be careful making strong conclusions about collar-specific plant species biomass variation.

Response: Above, we have explained how biomass sampling locations were selected in general. Unfortunately, we do not have any soil edaphic data but in locations were the samples were collected, the samples represented the vegetation in the experimental area based on visual estimation. Regarding competition, we refer to the height estimates of vascular plants, which were considered when choosing the sampling locations (Table A1).

With mosses, we prioritized sample purity (i.e., the sample consisted of the target species) and this aim dictated sampling locations and we did not examine any further plant traits. We acknowledge the concern that this may affect the reliability of estimating the biomass ratio of vascular plants and bryophytes and have revised the manuscript accordingly. Although this ratio is not an exact measure, we still think it is a robust indicator of community structure which can be used for e.g. modeling and upscaling purposes.

d) What do you mean by the "fertile" and "sterile" categories for the plant biomass? In my understanding fertile vs sterile categories are used in the context of evolutionary plant biology and plant reproduction (i.e. fertile vs sterile flowers). Or did you use it to somehow determine whether the species had vegetative culms (e.g. for *Carex*) from previous year? It is unclear to me how this classification is relevant to the topic of methane flux spatial variability, especially because you do

not talk about these classes afterwards. If you used some kind of scaling for the collar biomass (see comment 1 a), did you take use of these fertile/sterile classes in that as well? Please add a clarification for this separation, what the rationale is behind it, and what you mean by the terms.

It is now also unclear whether the species-specific plant biomass is calculated as the mean of n=20 biomass samples (fertile + sterile) per species per plot, or are the species-specific plant biomasses actually still divided into the fertile (mean of n=10 biomass samples) and sterile (mean of n=10 biomass samples) classes. Based on your results, it seems that you took the mean of 20 samples by combining the fertile and sterile samples? Please add a clarification to your methods.

Response: We have changed the wording from fertile to flowering and sterile to non-flowering. This division was done as the flowering shoots may have higher biomass than the non-flowering shoots. Although this division is not mentioned in the further text, it is used in the biomass estimations – flowering shoots (e.g. n=3 shoots) were first given the mean biomass of the flowering samples (e.g. 3 shoots  $\times 0.02$  g/shoot = 0.06 g) and the non-flowering shoots (e.g. n=5 shoots) a mean biomass of non-flowering samples (e.g.  $5 \times 0.01$  g/shoot = 0.05 g). Only after this, the biomasses were combined (0.06 g + 0.05 g = 0.11 g) and used in analyses.

Generally, we have revised the text in methods sections according to comments 1 a-d.

2. Since you are examining the spatial *variation* of methane fluxes within one study site and how plant biomass contributes to this variation (included in your research questions), I would suggest including additional measures of spatial methane flux variability (e.g., daily or seasonal coefficient of variation or other spatial variation metrics). This way you could quantify the spatial heterogeneity in methane fluxes which I think is currently lacking in this manuscript. Quantifying the spatial variation would be important background information for showing that there is indeed spatial methane flux variation in your peatland and then go to investigating the contribution of vegetation on it. You already touch on it a bit in 3.1 but only by talking about ranges in mean methane fluxes and visually showing plot-scale variability in Fig. 3 (which are good to show as well but do not really quantify the variation).

**Response:**

We have calculated a CV value for each day that measurements were recorded (19.10.2021-31.10.2022) and the daily CV ranged from 38.9 % to 300.4 % when looking at the whole year. When focusing on snow-free season, the range was 38.9 % - 85.4 %, and for snow season 39.3 % - 300.4 %. The largest differences were mostly from the time of the observed spring burst (1.4.-12.5.2022). We have revised the results section accordingly.

3. Did you measure methane fluxes from one plot once a day or multiple times a day? How did you decide which plots to measure each day when half of the plots were measured per day (n=18 out of n=36)? Please add more detail about this in the methods.

Response: Only one measurement was conducted per plot per day. On most days only half of the plots were measured, using randomized plot selection. We have revised methods section.

4. 116: The accumulated flux: it is based on a "24-hour" accumulated flux, but, based on your methods, you measured only between 8 am and 6 pm. It would be good to discuss that these accumulated fluxes are based on daytime fluxes and do not include nighttime fluxes, which may

lead to the annual accumulated fluxes being quite uncertain. You do mention that you assumed that the fluxes did not vary significantly in the diurnal scale but some justification may be needed here (can you refer to, e.g., the EC data to show this?).

Response: The assumption that fluxes did not vary remarkably diurnally was based on unpublished automatic chamber (AC) data from June 2022 measuring fluxes at the experimental site throughout the day. The AC data showed daily variation ranging from 0.27 mg/h/m² to 2.14 mg/h/m² during the month. These observations are also in line with the reference provided (Knox et al. 2021) in that way, that the site most similar to our study site (Lompolonjänkkä, FI-Lom) showed very little multiday or diel variation.

5. 118: you mention that you used the value from a previous measurement for days without flux measurements. What was the maximum number of consecutive days where there were no measurements? Methane fluxes (and some plant-mediated methane transport proxies) have been found to vary a lot in daily and multiday scales (see e.g. Knox et al. 2021: https://doi.org/10.1111/gcb.15661).

Response: The longest gap between the measurements was 28 days between 14.12.2021 and 10.1.2022. Daily mean flux on 13.12.2021 was 2.59 mg/h/m2 and on 11.1.2022 0.27 mg/h/m2. In comparison, the flux on 13.1.2022 was 3.61 mg/h/m2. The value used for these 28 gap days (2.59 mg/h/m2) represents the mean winter fluxes quite well (mean flux between 1.11.2021-30.4.2022 was 2.21 mg/h/m2). Other longer gaps between the measurements were 7 days (n=1), 6 days (n=3) and 5 days (n=3). We have clarified this in the manuscript.

6. 161-162: How did you test the significance between BM variables and methane fluxes using LOESS? LOESS does not test hypotheses, it is used for exploring nonlinear trends (which I believe you did here). You could rephrase this to highlight that (exploring nonlinear trends between BM and methane fluxes between VP clusters).

Response: We have rephrased the text accordingly.

7. 185 (figure 3). It is hard to identify the individual plots based on color in this plot. If they just represent the different plots without considering the vegetation composition within the plots, I don't think the coloring here is needed. You could just replace it with black lines and remove the legend (but mention that the black lines represent the different plots in the caption), for example.

Also, the plot numbers themselves do not really give any valuable information for the reader. If you want to show all the plot fluxes separately here, the plot numbers could be replaced with something simpler, such as 1-36, to improve the readability of this figure (the current numbering adds more complexity for the reader who is not familiar with your study site).

On the other hand, if you want to keep the coloring, could you do it based on e.g. vegetation composition grouping (for example based on your vegetation clusters)? If you do this, I would also recommend changing the red color of the soil temperature to black, because the red and green are difficult to separate visually for some readers with color-blindness.

The axis texts are also a bit small so please increase the font size.

Response: We have modified Figure 3 as suggested by the reviewer. We chose to combine the content of original figures 3 and 4.

8. 260-263: Could you add a sentence or two about how you would estimate the methane fluxes to change if you had measured them the same way as Alm et al 1999 or similar studies?

Response: Alm et al. (1999) observed that methane concentrations declined linearly within a snow depth profile towards snow surface. This indicates free upward diffusion through the snow and therefore, we would not expect highly changing flux rates even if fluxes were measured on the ground. The measurement would have been more accurate, though, and we could have linked it with the studied plant communities with higher confidence. We have added discussion about this topic.

9. 267-268 (and forward in this section): the plot numbers are not informative for the reader. You could instead describe the dominant vegetation of these plots (e.g. based on the vegetation clusters, which you show in B7)

Response: Revised.

10. 289-291: I understand your reasoning for the uncertainties in the wintertime methane fluxes and plant contributions on them but I would like to see some more in-depth discussion about this based on other studies. How can you make this conclusion based on your data? Depending on the snowpack properties (e.g. porosity) you might have also measured lateral methane flux which did not originate from the actual collar, especially in windy conditions or due to pressure changes between the chamber enclosure and the surrounding atmosphere and snow. Also, I would like to see a better reasoning behind your statement of the plants contributing to the measured winter methane flux even through the snowpack. In theory, this might be possible if there were broken stems or culms that were exposed to the air above the snowpack, which could possibly contribute to the Venturi effect via pressure changes especially in windy conditions but otherwise I am not currently very convinced, especially since you did not find any significant differences between the vegetation clusters in snow cover seasons (Fig. 5).

Response: We understand the critique behind this comment and acknowledge the uncertainties related to our data. However, regardless of the uncertainties, we found a significant correlation

between vascular plant clusters and wintertime methane fluxes in multivariate analyses (DCA and CCA) and, therefore, we argue that vegetation may play a role, and this topic should be acknowledged and potentially studied further.

We further justify our reasoning by referring in the revised version to previous work from other sites and observations from our study site. Pirk et al. (2016) observed that CH4 flux rates are, firstly, measurable, and secondly, do not decrease during the cold season, which indicates continuous CH4 production instead of a release of gas reservoirs. They observed a large spatial flux variability in the cold season and discussed that plant species composition, by affecting substrate quality and quantity, could cause these observed differences. In addition, they also observed in some plots a higher gas concentration at snow layers, where plant shoots located. (https://onlinelibrary.wiley.com/doi/abs/10.1002/2016JG003486). In addition, approximately 40 % of *C. rostrata* and *C. lasiocarpa* shoots at our study site overwinter green (Cunow et al., *in prep*). Therefore, it is plausible that gases travel through the aerenchymatous tissues of overwintering shoots, although this topic would require further examination. We have revised discussion accordingly with applicable references.

11. 292-307: I would move a majority of this part to the results section and discuss only the general aspects. Based on your research questions which are about the relationship between vegetation and methane flux, it doesn't seem so relevant to me to discuss the species distribution in such length here. This information would also be more useful in the results section, because then the reader has more of an idea about what kind of vegetation the individual plots contain and where they were located (see my previous comments about plot numbering, possible grouping in figures and naming).

Response: Text in second paragraph of section 4.2 partly moved to the results section 3.2.

12. 321: does it really provide labile carbon also in winter under the snowpack? Photosynthesis (and thus root exudation) is unlikely that efficient in those conditions, at least to the same extent as in the growing season, and especially so far up north as Puukkosuo where daylight hours are very few. Root decomposition could be one way too (but how efficient is microbial decomposition under the snow in cooler temperatures?) but I would like to see a bit more discussion based on more studies here.

Response: The assumption that *C. rostrata* "supports methane production year-round" was not related to photosynthesis. Cunow et al. (*in prep*) have studied the belowground processes and phenology of sedges at our site and discovered that approximately 40 % of *C. rostrata* shoots overwinter green. It is also likely that the roots of *C. rostrata* grow deeper than soil freezing depth. These observations support the assumption that the shoots could potentially act as conduits for methane produced in unfrozen peat layers during the cold season, transporting it through the frozen peat layers and snowpack. In addition, methanogenic bacteria have substrates for methane production also during winter and the substrate sources - fresh root litter vs. carbon stocks provided by the perennial sedges - likely depend on plant species. As noted by Pirk et al. (2016), microbial processes continue in the soil throughout the year, even though flux rates in the cold season are much smaller compared to the growing season.

(https://onlinelibrary.wiley.com/doi/abs/10.1002/2016JG003486. Given that the soil temperature at 5 cm depth at our site fluctuates around zero during the winter (Fig. 3) it is likely that methanogenic bacteria are also active.

**We have revised the discussion section.**

13. 342: it might be good to briefly discuss another explanation where vegetation would not be the main driver of the pH changes, and also add a bit more detail into how vegetation actually could have explained the pH variation.

Response: We did not claim the pH variation to be due to vegetation, but rather the opposite; the variation of pH likely explains the distribution of plant communities through bryophyte appearance. In the text, we note that vegetation composition was the primary driver of spatial variability of methane fluxes, and the significant relationship between these fluxes and pH was potentially explained through vegetation.

Based on the comments of the other reviewer, we have explored the relationship between methane fluxes and environmental factors a bit further and added some information about the role of pH in the text in section 4.4.

14. 344-346: This is an interesting finding and warrants a more detailed discussion. What do the other studies say and how could these theories apply to your study? Could the higher  $NO_3^-$  and  $NO_2^-$  concentrations contribute to pH or vegetation in some way that would enhance methanogenesis?

Response: We have revised discussion, highlighting the debated role of nitrate and nitrite in methanogenesis. We also added a finding that the positive correlation was not found when analyzing the environmental variables and methane fluxes in plots with no *C. rostrata*.

15. 349: this could indeed be the case but, to support this argument, you could also add a number to represent the lack of strong temporal variation in WTD (e.g. standard deviation or coefficient of variation if you want to compare growing vs non-growing season variation for example).

Response: We have WTD data only from the growing season, so comparing the values throughout the year is not possible. However, we have added the number of standard deviations in the revised version.

**16. 350-353: Two points:**

- a) The correlation between peat depth and plant biomass makes sense in the biological sense that, when there is more peat, there is also more space for roots especially for more deeply-rooting vascular plants. Since you did not find significant correlations between peat depth and methane fluxes, I would be careful drawing strong conclusions about the influence of peat depth on methane fluxes via vegetation (but see my next point).
- b) On the other hand, it is also possible that in the presence of deeply-rooted aerenchymatous vegetation, such as *C. rostrata*, the roots may provide labile carbon substrates in deep peat where methanogenesis increases despite the dominance of recalcitrant peat (i.e. indirect influences of peat depth on methane fluxes). The release of labile carbon compounds via root exudation could also trigger microbial carbon priming (see e.g. Waldo et al 2019: https://doi.org/10.1007/s10533-019-00600-6). However, be careful about your interpretations about the wintertime vegetation

influences based on your data (see previous comment about wintertime fluxes), and keep in mind that the direct relationship between peat depth and methane flux was still nonsignificant.

Response: We wanted to state that there was a significant relationship between peat layer thickness and the first ordination axes of vascular plant data in CCA and a positive pairwise correlation between peat layer depth and the biomass of vascular plants, sedge and *C. rostrata*, and that these latter three were all proxies for higher methane fluxes. This suggests, firstly, that these vegetation parameters associate with deeper peat and, secondly, may support methanogenesis through indirect relationships. We have revised the manuscript for clarity.

17. 354: Please add more discussion about why soil temperature may not have correlated significantly with methane fluxes- soil temperature has been an important predictor of methane fluxes in multiple studies and discussing this opposing result would be warranted.

Response: Olefeldt et al. (2017) have found that methane fluxes in boreal fens are associated with soil temperatures at greater depths rather than with surface temperatures. As we only recorded temperatures which reflect the surface temperature, our results are in line with the Olefeldt et al. (2017). We have discussed the lack of correlation between soil temperature and methane fluxes in discussion.

**Technical comments:**

- 30: Add "(CH4)" after "methane". You could also replace the rest of the "methane"s with "CH4" if you want, especially since you use it in the flux units throughout the paper.
  - Response: Corrected. In text we prefer using the word "methane", as it is commonly read out as a word instead of the chemical formula 'CH4'.
- The word "dynamics" is used quite a lot throughout the introduction. I would recommend changing it to something more specific, as in some cases (e.g. "methane dynamics") it may sound a bit vague.
  - Response: We have replaced word "dynamics" with more specific terms throughout the text.
- Generally through the whole manuscript: the term "year-round" doesn't sound very good to my ear. How about "full-year"?
  - Response: We have carefully chosen the phrase "year-round" because it clearly expresses the nature of our data continuous, frequent measurements across an entire year. This term highlights that the data is not aggregated into an annual sum but rather presented as a time series of spatial variation. Therefore, we believe "year-round" is the most appropriate term and prefer to retain our choice of wording.
- 32: Saunois et al have a newer global methane budget paper (currently a preprint): https://doi.org/10.5194/essd-2024-115
  - o Response: Reference updated.
- 34: instead of using the word "spatial and temporal dynamics", maybe "spatiotemporal variation" or something similar would be better?
  - Response: Corrected with "spatiotemporal patterns".

- 39: "ecosystem process" maybe use another word, for example "These ecosystem-level processes..."
  - o Response: Corrected.
- 42: remove "layers" after topsoil, and add why rising temperatures lead to increased topsoil oxidation?
  - Response: It is probable that increased temperatures enhance microbial activity and oxygen availability (Zhang et al. 2021). We have removed "layers" and added this clarification.
- 45: I would change the topic sentence to something shorter. Perhaps remove mention of hydrology and just start with "Vegetation type and its responses.."
  - o Response: Corrected.
- 47-48: maybe change the words "deeper" and "upper" to "anoxic" and "oxic" (this way it would focus on methane being transported from anoxic soil through the oxic soil and into the atmosphere)
  - o Response: Corrected.
- 49: I would be careful with the wording "better than any abiotic factor"- please add more references, or modify the sentence so that it doesn't sound so definitive
  - Response: Edited the sentence to "Indeed, plant species and their specific traits have been found to be **reliable** predictors of methane flux rates (Korrensalo et al., 2022)."
- 59: the part "extensive, year-round, plot-scale flux data are, however, limited" sounds a bit complicated. Maybe something like "However, ... full-year methane flux data at the plot scale are limited"?
  - o Response: We understand it is a complex sentence and will revise it.
- Methods: the model numbers could be written in parentheses after the instrument, e.g. at row 99 you could put the LI-COR model number "LI-7810" in parentheses after mentioning the instrument.
   You already do this in the 2.5 section so it would be good to keep it consistent.
  - o Response: Corrected.
- 60: would "..spatial variability in methane fluxes." work better?
  - o Response: Corrected.
- 71: what is "normal period"?
  - Response: A "normal period" refers to a climatological standard normal. This is a period used to calculate average climate conditions, spanning over 30 years. The 30-year period used for calculations is stated in row 71.
- 73-74: please add a detail saying where the pH was measured (peat I assume?)
  - Response: pH was measured from peat pore-water, which is mentioned in the revised text.
- 76: was the variation standard deviation or other measure of variation? Or do you mean that 6.3 cm was the mean WTD during the study period? Please specify.
  - Response: The plot-scale variation in WTD during the snow-free season of 2022 was 3.8–9.1
     cm with an average of 6.5 cm. This is corrected in the revised version.

- 77: graminoids are herbaceous plants so this sentence should be changed accordingly (you could, for example just call them "vascular plants typical of rich fens" and then give the species examples)
  - o Response: Corrected.
- 91: The figure caption could be made even simpler, how about just starting from: "A map of Puukkosuo rich fen.."?
  - o Response: Corrected.
- 95: You could remove the mention of "manual" here since you introduce it later in this paragraph.
  - o Response: Corrected.
- 97: this sentence ("..., doing measurements from half (n=18) of the study plots per day") could be made smoother, for example just: ".. from half (n=18) of the study plots per day".
  - o Response: Corrected.
- 105: the end of the sentence starting with "making the possible dilution.." is a bit hard to understand, could you make this a bit clearer? Do you mean leakage? Good that you mention this though.
  - Response: We purely meant any effects that the snowpack may have on the magnitude of the flux, without specifying exactly what they could be. We have revised the text accordingly. The uncertainties related to this measurement technique are further discussed in section 4.1
- 108: what exactly do you mean by "successful"? Visible linear increase in CH4 concentration?
  - Response: We have revised the text to more clearly state that we only accepted the measurements with an R² value ≥ 0.95 (n = 3589) and inspected all the rest (n = 691) individually, leaving out measurements showing very strong non-linearity or any other sign of failed measurement (n = 159).
- 115: it is a bit unclear now how you determined the snow cover- did you define it snow-free when
  there was snow but you were able to set the chamber on the collar? For transparency, it might be
  good to add this detail here.
  - Response: We defined the seasons by the ability to measure the fluxes of all the 36 plots on the collar. We have revised the text accordingly.
- 145: do you have more details of the pH analyzer, other than the brand?
  - Response: Details for the pH analyzer (913 pH/DO Meter, Metrohm) were added to the text.
- 146-147: write the numbers in the molecules in subscript (e.g. NH4)
  - Response: Corrected\_throughout the manuscript.
- 148: please add that you estimated the litter cover as a separate percentage cover, if this is the
  case. This could also be actually mentioned already in the plant community data where you talk
  about moss percentage cover.
  - Response: We consider litter cover as a separate environmental variable, not included in the plant community data, and would rather keep the mention of it only under section 2.5.
     We have edited the text to clarify the estimation process.

- 151: "VP" abbreviation appears here for the first time but you don't introduce it before this. Please add the abbreviation to the appropriate spot in the text (maybe introduction?) so you can then start using it: "vascular plants (VP).."
  - Response: The abbreviation is introduced earlier in section 2.4. We will inspect the consistency of the use of abbreviations.
- 181: put the "4" in "CH4" in subscript
  - Response: Corrected.
- 204: add the name of the statistical test you used to obtain the F-values ("F=..") for the first occurrence of the letter.
  - Response: The name of the statistical test (ANOVA) added.
- 206 and forward: write the species names in italics and I would also write the complete names, e.g. "C. rostrata". It would improve the readability if you wrote them in full form (the genus does not have to be written out since you have already discussed the species before).
  - Response: The original idea was to separate individual species from the plant community clusters indicated by these species. We understand that this might have caused unclearness in the text and have therefore corrected the unclarity related to this issue starting from section 3.2, where the clusters are first described.
- 215 (figure 4): please increase the axis text font size, and consider writing out the species names in the legend. Caption: replace "dot in the graph" with "data point". Based on this plot, it also seems that there might be another plot group or cluster in T. ces where there are lower fluxes (the lower yellow point cloud which I would imagine could lead to a different smooth curve? Did you look into this? What might contribute to this trend? This is a bit extra but maybe worth discussing and/or looking into.
  - Response: We have updated the figure, and the caption based on the reviewer's comments.
     We chose to combine figures 3 and 4 into one graph. While the plot gives an impression of a possible fourth cluster showing lower fluxes during the peak season, the cluster analysis for vascular plants didn't imply this. Please, see cluster dendrogram shown in Figure B2.
- 220 (figure 5): 1. increase the font size for axis texts. 2. Even though you list them in the caption, I would still write out the whole species names instead of the abbreviations in the plot. 3. It is very hard to see the median line in the dark blue boxplots so changing the color to something lighter might help readability. Also, even though the colors look nice, are they really needed here since you also give the same information on the x axis as cluster names? Or, if you would prefer keeping the colors, you could also consider removing the legend and in the caption write something along the lines of "the colors represent the clusters and are shown for clearer visualization".
  - Response: Figure edited mostly as suggested. To keep the figure clear, we chose to use the abbreviations of the species names in the plot instead of full names.
- 225: to remind the reader what these are, please add the term before "DCA" and "CCA" abbreviations: e.g. "detrended correspondence analysis (DCA)".
  - o Response: Corrected.
- 226 forward: you could write the species names in complete forms here.

- Response: Corrected. We want to emphasize that we have explained earlier in the text that
  these cluster names refer to plant communities indicated by these species, rather than
  purely the abundancy of these individual species.
- 243: would something a bit more specific be better instead of calling the ratio "BM ratio"? For example, "VP:BRYO ratio"? The reader might forget what exactly "BM ratio" consists of and would need to come back to the definition of this term.
  - o Response: We have revised the text and the abbreviation in Figure 5 (scatterplot).
- 245: add "p" to the second p-value: "and p ≤ 0.01".
  - o Response: Corrected.
- 250 (figure 6): increase the font size of axis texts and write out the species names in italics.
  - Response: Figure edited partly as suggested. Additional scatterplots added to the figure to further show the significant biomass and environmental variables affecting snow-free season methane fluxes.
- 257: move the Jammet et al reference to the end of the sentence, and if possible, try to find another reference here since you mention multiple northern fens. Or you could just say ".. in a northern boreal rich fen (Jammet et al. 2017)."
  - Response: We chose to refer only to Jammet et al. (2017) as it was from a site most similar to ours.
- 256-260: I think these sentences should be in the results section and not in discussion. For example in the 3.1 section.
  - Response: Part of the text from beginning of section 4.1 moved to section 3.1.
- 264: is this percentage based on your results? If yes, please indicate so in this sentence, and if not, add a reference.
  - Response: The percentage is based on our findings (2.3-21.3 %) and the findings of Alm et al. (1999) (6-17 %) and this has been clarified in the revision text.
- 274-275: plant traits are part of vegetation, so you could rephrase this by for example: "...could not be explained by aboveground plant biomass.". Also, give examples of these plant traits, as well as the "ecohydrological aspects" and microbiota, and how they might contribute to the spatial variability in CH4 flux between the plots.
  - Response: We have corrected the first point as suggested (row 306). Also examples about plant traits, such as rooting characteristics, ecohydrology, such as peat water holding capacity, and microbial metabolic interactions, such as nutrient cycling, as well as their contribution to spatial variation in the fluxes (contribution to soil conditions, substrate availability, and microbial activity) have been added to the text.
- 315-316: replace the "organic matter" with "carbon substrates", and replace "and providing pathways" for example with: ".. for methanogenesis through deep root systems throughout the year".
  - o Response: Corrected.

- 318: add "methane" in front of "transport": "..may be due to the species' high methane transport rate.."
  - Response: Corrected.
- 318-319: why would *C. rostrata* have low oxidation potential in your study? As you say next in this sentence, this species has high root porosity (so it could also oxidize the rhizosphere), so why would the methane transport exceed the effect of methane oxidation in your study? Clarify briefly.
  - Response: The assumption is based on a previous study by Ström et al. (2005) where *C. rostrata* was observed to have a much lower capacity (20-40 %) to oxidize the rhizosphere compared to two other species (*Eriophorum vaginatum* and *Juncus effusus*) (>90%). We have edited the text to highlight the contrast between the plant traits and low oxidation potential.
- 319: saying both "high porosity" and "large aerenchyma" is not needed as they refer to the same thing. You could instead just say ".. and high root porosity".
  - o Response: Corrected.
- 325: this is a bit vague sentence. How about: "Thus, VP:Bryophyte ratio could be used as a parameter in peatland methane flux models together with remotely-sensed data products." (But see comment 1 c)
  - o Response: Corrected.
- 330: add "gas" "high transport efficiency": "high gas transport efficiency"
  - Response: Corrected.
- 332-333: This sentence is a bit unclear. Do you mean that *C. rostrata* had more shoots and therefore plots with more *C. rostrata* shoots transported and emitted more methane?
  - Response: Due to the species' high transport efficiency, even a few shoots can release the total flux magnitude from the ground. We observed saturation in the magnitude of the flux, rather than a linear increase with higher biomass of the species. This indicates that only a few shoots of *C. rostrata* are needed to release the methane stored in the soil. We have revised the text.
- 334: add "methane" to "transport efficiency": "methane transport efficiency"
  - o Response: Corrected.
- Figure 7: move this to the results? And increase the axis text font size and write out the species names in italics.
  - o Response: Corrected.
- 359: remove the mention of "causality" because you did not use methods for estimating causal relationships in this study.
  - o Response: Corrected.
- 361: remove "answer our first research question and", and replace "affects the flux" with "affects methane flux".
  - o Response: Corrected.

- 363-364: remove "answer our second research question and"
  - o Response: Corrected.
- 368: you didn't really discuss plant traits in the discussion part, so I would remove the mention of plant traits here. Or, you could say for example: "Our findings suggest that, in addition to species-specific plant traits, the biomass ratio of vascular plants and bryophytes could potentially be used as a parameter for predicting peatland methane emissions" (but see comment 1 c).
  - o Response: Corrected.
- 373: I don't think you have to show the reference at the end. The closing sentence would be stronger without it.
  - o Response: Corrected.

**REVIEW COMMENTS #2**

This manuscript presents results from a study of methane fluxes in a northern calcareous fen and investigates the role of plant community composition on the measured fluxes. Methane flux was measured throughout the annual cycle, allowing for an investigation of the role of vegetation in both snow-free and snow covered periods. As little data is available in the wintertime in northern peatlands, this adds to our understanding of winter and annual methane emissions in peatlands.

Overall, the study is careful conducted, and an impressive number of methane fluxes were measured and used in the analysis. Plots were then assigned to plant community types based on a cluster analysis of species composition and the presence and biomass of Carex rostrata was observed to result in higher methane fluxes in the snow-free period and annually. Bryophyte composition alone was not a good predictor of spatial variation in methane flux. Although the authors indicate that other environmental variable such at water table depth (WTD) were not strong predictors of methane flux, it is possible that these interacted with the plant community, although this was not fully explored in the present analysis. Currently, the role of environmental variables is largely assessed with a correlation analysis, but I provide some additional suggestions of how this could be further explored. Given the wet nature of the study site, it still may be that these environmental variables do not explain much variation. However, there is a large amount of unexplained variation in methane flux among plots where sedge biomass is low, and it may be that WTD or soil temperature (or some of the other variables measured) would explain some of this variation if these plots are investigated separately.

My other main suggestion is to improve clarity in some of the methods and reporting of results, with specific suggestions outlined below.

Response: We thank for the thorough feedback on our manuscript. We want to highlight that multivariate analyses were performed to the plant community data and that the correlation values are achieved both from Canonical and Detrended Correspondence Analyses (CCA and DCA), and pairwise comparisons with Pearsons' correlations. We have clarified this throughout the manuscript to show which correlations are from which analyses. Further comments related to this topic can be found from the following responses.

**Specific comments:**

Lines 21-23: It was not clear to me exactly what this sentence was aiming to convey. Can you reword to make this clearer. I have tried to interpret it and if the following suggestion captures the correct meaning, then you can use it to help with the update "Plant community dominated by Carex rostrata

accounted for 13 of the measured plots with these plots contributing 44–49% of the measured methane flux during the three periods".

Response: What we actually meant was that *C. rostrata* was present (as an individual species) at 13 plots. Only 10 plots were clustered as *C. rostrata*-community clusters, but the species was also present in other clusters. With this sentence we wanted to highlight the contribution of the species to high fluxes, not only the community clusters indicated by the species. We have edited the sentence as follows: "*C. rostrata* was present at 13 out of 36 plots, and these 13 plots contributed 44–49 % of the measured methane fluxes during the three periods."

Line 30: northern peatlands are not the main terrestrial source of methane. Wetlands may be, but northern peatlands only account for a small portion of the wetland total. Please update this sentence for clarity. e.g., the global methane budget indicates wetland emissions of 248 Tg CH4/yr https://www.globalcarbonproject.org/methanebudget/, while the reference cited here estimates emissions of only 38 Tg CH4-C/yr from the whole northern region.

Response: Corrected.

Line 50: oxidize instead of oxidate

Response: Corrected.

Line 131: How did you cut them to estimate biomass? Did you include only green tissue, or some depth that you considered active? As am sure you know, it can be difficult to define living moss biomass, so a few more methological details would be useful here.

Response: We removed the non-living parts based on subjective assessment which was consistent between the samples (the first authors was solely responsible of this). We included only the colorful or leafy parts to represent the aboveground biomass. We have revised the manuscript accordingly.

Lines 137-138: What was the extent of microtopography at the site? Did you consider correcting the WTD measurements for local elevation variation to better represent the actual WTD at the flux measurement collars? Do this effect the interpretation of the role of WTD for accounting for variation in methane flux?

Response: The experimental site is designed so that majority of the experimental plots (n=30/36) are located in lawn microhabitats that also generally depict our study site. The remaining six plots are located in mud microhabitat. Based on personal observations, hollows and hummocks do occur at the site. Small scale variations in microtopography do occurr but generally the variation is relatively minor. The suggestion to correct the WTD for local elevation could have increased the accuracy of the measurements but, unfortunately, this was not considered at the time of fieldwork. However, the WTD throughout the fen during snow-free season varied only approximately 15 cm (between -9.3 cm to 7 cm) and in plot-scale, the averages of minimum and maximum values were -4.7 cm and 1.8 cm, respectively. Therefore, we think it is reasonable to assume that the WTD did not vary extremely within the 1 m distance, i.e., the distance between the collar and the WTD tube.

Line 164: It's not clear if this VP to bryophyte ratio is based on biomass or cover. I assume biomass, but it can be made clearer in the text.

Response: The ratio is calculated as biomass of vascular plants divided by the biomass of bryophytes. We have clarified this in the text.

Line 165: Please provide additional information about the correlation analysis. Was this Pearson correlation? It also isn't really clear that this was done using average conditions across the sample periods and not instantaneous values (this is my interpretation after reviewing Table B2), which would likely give different results (so just be clear here in the methods what was done). Also, see my comments below about considering the variables together in a multiple regression analysis to assess whether there are interactions between the environmental variables and vegetation clusters.

Response: The correlations values in Table B2 are Pearson correlations. However, we gained correlation values also from multivariate analyses (DCA and CCA). In the revised manuscript we clearly state from which analyses the correlation values are derived from. In the analyses we used snow-free season averages to represent the overall environmental conditions since parameters were measured at different frequencies. We think that this instantaneous values approach provides a comprehensive view of long-term trends and reduces short-term variability, aligning with our study objectives and the multivariate analysis approach.

Lines 193-194: Are the numbers in brackets averages across all the plots? Please specify in the text and include an estimate of variation (e.g., standard error or standard deviation)

Response: The numbers are total biomass estimates of all the plots, calculated with the biomasses of the collected samples. We have clarified this in the text but do not find it necessary to add standard deviation for the total biomass, as it was not taken into consideration in any of the analyses. Biomasses and the standard deviations of the collected samples can be found in Table A1.

Lines 199-200: I had a hard time following which species were with which cluster. Maybe add numbers in front of the clusters to clearly separate them.

Response: The other reviewer also criticized this and therefore, we now use only the name of the strongest indicator species as a name for the cluster and revised the text accordingly throughout the revised manuscript.

Lines 264-266: I totally agree with this statement, but maybe it is also important to highlight here that fluxes were much less variable over the snow-period and did not vary significantly among the identified species clusters. This would suggest that even much lower sampling effort could effectively capture winter fluxes, helping to estimate annual emissions. Vargas and Le 2023, Biogeosciences also supports this conclusion <a href="https://doi.org/10.5194/bq-20-15-2023">https://doi.org/10.5194/bq-20-15-2023</a>

Response: As the reviewer is suggesting, our observations are in line with Vargas and Le (2023) that lower sampling efforts during the mid-winter could effectively capture the fluxes, aiding in the estimation of annual emissions. We have added this in the revised manuscript.

Lines 275-280: What about interactions between the soil environment (WTD and temperature) and the plant communities? You didn't find a significant correlation of CH4 with WTD alone, but it could be that there was a significant correlation in some plant communities and not other, resulting in no significant pattern across the whole dataset. Did you consider multiple regression models that included the plant community information alongside the environmental drivers?

Response: Please, note our response above where we describe the use of community analysis - both Detrended Correspondence Analysis (DCA) and Canonical Correspondence Analysis (CCA) - followed by Pearson correlation. CCA captures the complex interactions between the species, community clusters and all environmental variables used in the analysis. The plant community compositions we used were 1) All species (including both vascular plants and bryophytes), 2) Vascular plants alone, and 3) Bryophytes alone. In these analyses, pH and WTD were strongly connected to bryophyte clusters but

not to vascular plant clusters. Soil temperature at 5 cm depth did not show significant correlation with any community cluster. Therefore, it is correct to state that some communities correlate significantly with edaphic factors and these relations are discussed in detail in sections 4.2 and 4.4.

Lines 289-291: This does not seem to align with the results in Figure 5a where there were no significant differences in CH4 flux among the VP clusters in the snow-period. Are you overstating here?

Response: In figure 5a we show linear regression, which does not show significant differences between the communities, like the reviewer noted. However, in multivariate analyses, the first ordination axis of vascular plant data correlated significantly with snow season fluxes (r = 0.445 in DCA, -0.402 in CCA). The significant results during the snow season from DCA and CCA but not from linear regression models suggest that the relationship between vegetation and snow season methane fluxes is complex and nonlinear and likely involves interactions with other environmental factors that linear regression did not capture. CCA's ability to handle multivariate data provides a more reliable and accurate representation of the ecological dynamics during the snow season in comparison to linear regression models. This explanation has been added to the revised version.

Line 325: I'm not sure how easy it will be to estimate BM ratio with remote sensing. The community identity or importance of sedges on an areal coverage basis would seem like a variable that is easier to measure with imagery.

Response: With remote sensing we mean mainly multispectral imagery, which has been shown to detect ratios of vascular plants and mosses reliably (Wolff et al. (2023), <a href="https://www.sciencedirect.com/science/article/pii/S1470160X23002820">https://www.sciencedirect.com/science/article/pii/S1470160X23002820</a>). We have clarified this in the revised version.

Line 341-342: It's not clear to me where you did this? The correlation analysis looks at each variable individually and does not consider if they interact in predicting flux. Based on Figure 7, there is a wide range of CH4 fluxes when C. rostrata biomass is low, so some other variable must be explaining this. Is it possible that the response of CH4 to the environmental variable differed among the clusters? Did you investigate this (e.g., something like an ANCOVA or multiple regression with the cluster type as a categorical variable that interacts with things like pH, WTD and soil temperature)?

Response: The values in 4.4 are again from both multivariate analyses and Pearson correlations. The reviewer is right that there is variation in the fluxes that cannot be explained by C. rostrata biomass – an observation that we have discussed shortly in the manuscript. However, it was an interesting idea to explore the difference in responses to environmental variables between the clusters still further. To address the reviewer's concern, we chose to examine the plots which did not have any C. rostrata growing in them (23/36 study plots). We analyzed the environmental variables together with snow-free season methane fluxes only on these plots to see which other factors, in addition to C. rostrata biomass, might show significant relationship with the fluxes. We tested this with linear model and ANOVA in R studio and discovered significant relationships between the fluxes and pH (p < 0.001) and litter coverage (p = 0.01). None of the other environmental or biomass variables resulted in a significant p-value. The Pearson correlation values between snow-free season methane fluxes and pH and litter were -0.66 and -0.29, respectively. These findings have been included in the revised manuscript in results and in discussion.

Line 377: I highly encourage the authors to deposit the full datasets in an open access data repository to ensure availability of the data for future studies/meta analyses.

Response: We agree on the importance of open access data. However, data used for this study is part of a larger dataset of EcoClimate experiment at Oulanka Research Station. Full data repository is being developed, and all the data will be freely available at some point in the near future. You can familiarize yourself with the experiment via these links:

https://www.oulu.fi/en/research/research-infrastructures/oulanka-research-station

 $\frac{\text{https://anaee.fi/facility/ecoclimate/\#:}^{\text{:text=Oulanka}\%20 research\%20 station\%20 owns\%20 and\%20 oper}{\text{ates}\%20 the}\%20 EcoClimate, instrumented\%20 and\%20 is \%20 designed\%20 to \%20 run\%20 for \%20 decades.}$

Table B3: Please add units to the columns.

Response: Corrected.

---

## Referee Report (RR1)

The revised manuscript by Järvi-Laturi et al. presents very valuable findings of vegetation-driven year-round methane flux variation in a northern fen. One of the greatest contributions to the field of methane research in this manuscript is the inclusion of non-growing-season methane flux data and the presented evidence of vegetation contribution to wintertime methane fluxes. I think these results are very interesting and the importance of these findings has been highlighted more clearly in the revised version. I believe that this paper will motivate more researchers to study wintertime methane fluxes and particularly the contribution of plants to them, an aspect that has been lacking for a long time.

I recommend accepting the revised manuscript as is.

---

## Referee Report (RR2)

**Plant community composition controls spatial variation in yearround methane fluxes in a boreal rich fen**

By Eeva Järvi-Laturi et al.

This referee report concerns the revised manuscript. The initial submission was reviewed by two referees; their detailed comments, together with the author responses, have been taken into consideration by this referee, following evaluation of the revised manuscript at 'face value'.

This is a robust and important year-round study of the relationships between vegetation community composition and methane (CH4) fluxes in a boreal rich fen; an ecosystem type for which such data are very scarce. The spatiotemporal variability of CH4 fluxes has been analysed and interpreted based upon 4121 hard-won individual measurements, using a manual closed-chamber approach over 36 study plots year-round. The revised manuscript reads well, and this is an important contribution to the field, highlighting the potential to upscale emission predictions and improve ecosystem-scale CH4 modelling by identifying vegetation-related emission hotspots.

The manuscript is very strong in its current form, although there are a few remaining aspects which might be worth further consideration to get the most out of this study:

On lines 116-20 the authors state that "Annual accumulated flux (1.11.2021-31.10.2022) was estimated by calculating a 24-hour accumulated flux for each available datapoint by multiplying the hourly mean flux by 24. These daily flux values were then summed to obtain the annual total. The days which were missing a measurement were given the value from a previous measurement, assuming the fluxes did not vary remarkably diurnally or over the days". The assumption that 'fluxes did not vary remarkably diurnally' requires further justification/consideration, in my view (also; should 'diel' replace 'diurnal' – check definitions?). The measurements were taken between 8 am and 6 pm, using a clear polycarbonate chamber. Thus, photosynthesis and stomatal conductance will likely reflect daylight conditions, with open stomata (in living tissues, during the snow-free season). Is it possible, therefore, that extrapolation from day-time measurements to 24-hour flux values could cause a systematic overestimation of daily flux rates, where aerenchymatous CH4 transport is important (I have provided web-links to some potentially relevant papers at the end)? If release of CH4 via the stomatal pathway (as opposed to via leaf micropores and/or the epidermis/cuticle) is potentially important then I think it is worth noting in the manuscript. If no 'around-the-clock' flux measurements are available from this site then this does not undermine the paper; rather, this issue should be noted and discussed. Indeed, based on the results presented here it

would be valuable, in any future study, to take some (snow-free season) 24-hour measurements, especially in plots belonging to the *Carex rostrata* cluster. Note that I am aware of the latitude of Puukkosuo fen (66.377299° N), and the implications for light climate.

Lines 65-66, 336-37 and 384-85 (the final sentence of the Discussion) state, respectively, "We hypothesize that (1) the plant community composition affects the methane flux ...", and "...plant functional type and species largely determine the magnitude of the fluxes" and "All these findings highlight that vegetation, rather than environmental factors, was the main driver of methane fluxes at our site." However, because plant community composition itself reflects (and interacts with) site physicochemical environmental factors, it is important to be very careful with the wording here, and assignment of 'cause and effect'. I would therefore urge the authors to reflect on this one more time, prior to final publication, and consider whether these statements remain robust and objective, or whether some caveats should be introduced. I am not disagreeing with these statements, but plant community composition is not independent of site-level environmental factors, which themselves may influence CH4 fluxes. Indeed, I wonder if the title of the paper could perhaps be amended (slightly!) to "Plant community composition explains spatial variation in year-round methane fluxes in a boreal rich fen"?

Related; lines 274-75 state that "There was no significant correlation between methane fluxes and WTD or soil temperature in any period." I found this remarkable, based on Fig. 3, which shows a broad relationship between soil temperature and CH4 fluxes for all vascular plant clusters on a seasonal basis. At the end of the Discussion section (lines 382-84), however, the authors explain that "methane fluxes did not correlate with peat temperature at 5 cm depth. Indeed, methane fluxes in boreal rich fens associate with deeper soil temperatures, which connect to water table position, rather than with surface temperatures influenced by air temperature (Olefeldt et al., 2017)." Had soil temperature data been available from deeper in the profile then do the authors consider that they might have been able to detect a relationship between temperature and CH4 flux; or is it solely, as they claim, that "vegetation, rather than environmental factors, was the main driver of methane fluxes at our site" (line 385)? Put another way, are the authors confident that this final statement, in the absence of relevant (deeper) soil temperature data, is robust?

Some more minor points for consideration:

Lines 52-53 - A very bold statement appears here, reliant upon just one reference: "Climate change is predicted to accelerate the natural vegetational succession in boreal rich fens towards Sphagnum-dominated plant communities even in stable hydrological conditions (Kolari et al., 2021)." I would therefore suggest modifying the sentence to "Climate change is predicted to accelerate the natural, autogenic, vegetational succession in boreal rich fens towards *Sphagnum*-dominated plant

communities, even in stable hydrological conditions (see Kolari et al. (2021), and references therein)."

Line 135 – Delete the comma, to read " ... for those species which were found flowering ..."

Line 150 - The units " g/1 %" appear, which in the manuscript font can look like g per litre. I therefore suggest writing this in full; i.e. g per 1%.

Caption of Figure B7 – correct the spelling of segregated (from segragated).

**References relating to CH4 transport through vascular plants:**

https://doi.org/10.1104/pp.94.1.59

https://doi.org/10.1016/0045-6535(93)90430-D

https://doi.org/10.1016/0304-3770(93)90040-4

https://doi.org/10.1016/0304-3770(96)01048-0

https://doi.org/10.1016/j.atmosenv.2003.09.066

https://doi.org/10.1016/j.aquabot.2004.10.003

https://doi.org/10.1046/j.1469-8137.1998.00210.x

https://doi.org/10.1111/j.1469-8137.2012.04303.x

https://doi.org/10.1007/s10533-019-00600-6

https://doi.org/10.1002/lno.11467

---

## Author Response (AR2)

**Report #1**

The revised manuscript by Järvi-Laturi et al. presents very valuable findings of vegetationdriven year-round methane flux variation in a northern fen. One of the greatest contributions to the field of methane research in this manuscript is the inclusion of non-growing-season methane flux data and the presented evidence of vegetation contribution to wintertime methane fluxes. I think these results are very interesting and the importance of these findings has been highlighted more clearly in the revised version. I believe that this paper will motivate more researchers to study wintertime methane fluxes and particularly the contribution of plants to them, an aspect that has been lacking for a long time. I recommend accepting the revised manuscript as is.

Response: We are happy to hear that you found our manuscript improved and thank you for appreciating our work.

**Report #2**

The authors have thoughtfully considered comments of two reviewers and thoroughly revised the manuscript to address the concerns raised. The clarifications that the authors have added to the methods and results have made it much easier to follow the findings and conclusions from the study. I recommend publication but suggest a few very minor revisions for further clarity in the final version.

Lines 118-120: In the response to reviewer 1, the authors provide several pieces of evidence to indicate that there is minimal diurnal variation. I suggest that reference to this information is provided here to support the stated assumption.

Response: Our assumption regarding limited diel variation is supported by continuous  $CH_4$  flux measurements from two automatic chambers located at the same site near our study plots (Mastepanov et al., unpublished data). Based on this dataset, we calculated the daily standard deviation of  $CH_4$  fluxes for each chamber during the peak flux season (July), when flux values ranged around 12 to 15 mg  $CH_4/m^2/h$  and diel variation is typically most pronounced. We found that the average daily standard deviation was 0.758 mg  $CH_4/m^2/h$  for Chamber 1 and 0.696 mg  $CH_4/m^2/h$  for Chamber 2, indicating limited diel variability. To compare, we also looked at the standard deviation for snow season fluxes and discovered standard deviation of 0.518 mg  $CH_4/m^2/h$  for Chamber 1 and 0.439 mg  $CH_4/m^2/h$  for Chamber 2. This clarification has been added to lines 124–125 in the manuscript: "These assumptions were based on data from two automatic chambers located at the same site near our study plots, which show limited diel variation in fluxes (Mastepanov et al., unpublished data)."

Line 159: I had previous inquired about microtopography at the site and the authors note in their response that microtopographic variation is minimal and that WTD measured 1 m from the plot likely provides a reasonable estimate of WTD at the plot. I suggest adding a sentence with that information here as other readers are likely to also question the representativeness of the WTD measurements.

Response: To clarify microtopographic variation we have added text on lines 158-159: "As the microtopographic variation and WTD fluctuation at the site is minimal (see 2.1), WTD measured 1 m from the plot likely provides a reasonable estimate of WTD at the plot."

**Report #3**

**Below you find my report:**

The authors present a one year study from a northern rich fen in which methane emission measurements were conducted on 36 plots in order to assess the influence of the vegetation community composition on methane flux. Further, the authors report that the biomass of vascular plant, sedges and especially C. rostrata and the ratio of biomass of vascular plant to biomass bryophytes correlated well with measured methane fluxes. Pointed out in particular is the strong influence of C. rostrata with regards to methane emissions during the snow free season. Here C. rostrata seems to have a large enhancing effect on methane emissions.

The study is based on an impressive number of flux measurements and a sophisticated spatial setup to cover potential spatial variability.

The research conducted is of high quality and I have no doubt that the approach is scientifically valid and robust, however I have some major criticisms that need to be assessed:

Response: We thank you for the positive feedback and the criticism. Below, we have addressed the comments one-by-one:

1. The conclusions drawn from the results are very strongly based on aboveground biomass. As only aboveground biomass was sampled firstly I think it is necessary to more thouroughly discuss the multitude of impacts the belowground biomass can have on the methane flux, be it negative or positive. Secondly, the aspect that the biomass was sampled outside the plots to which the methane fluxes were associated too adds to the uncertainty and this uncertainty should be stressed even more in the discussion as other abiotic factors may influence methane flux very strongly. These factors are also known to be highly variable in space and time (WTD, soil temperature, micro topography etc.) This is not a general criticism of the method, however, I think this needs to be stressed more strongly. Thus, the conclusions, although valid, should be slightly relativised in my opinion. With this regard I find it overstating that plant community composition controls the entire spatial variation of methane flux, as it is mainly an effect of C. rostrata. Please consider the special plant traits of C. rostrata in your discussion and consider changing the title slightly.

Response: Firstly, in our paper we concentrate on the spatial variation of the fluxes, rather than the functional aspects of it, like aboveground vs. belowground parameters. However, we acknowledge the importance of the belowground parts of the plants and agree that it is probably one of the main drivers of the spatial variation of methane fluxes. As a response to this comment, we have added two sentences related to this topic on lines 381–385: "This study focused only on aboveground parts of the plant communities, and therefore the multiple effects that the belowground parts may have on methane production, consumption, transport and emission (e.g. Määttä and Malhotra, 2024) were not considered in the analyses. However, we acknowledge that belowground plant characteristics likely play a significant role in explaining the variation in our methane flux data that could not be explained by the other studied variables."

Secondly, our study site is a sloping flow-through fen, which has rather homogenous microtopography, water level, and soil temperature (as partly explained in manuscript in section 2.1). Further, the locations of biomass sampling were chosen to reflect the height of the vegetation at the plots to lower

the uncertainty of the sampling (see lines 137-138). Due to these reasons, we believe that the biomass samples reflect the biomass of the plots strongly enough for making these conclusions.

Thirdly, it was, in fact, the point of this paper to talk about species as an indicator of a plant community, and to show that spatial variation of methane fluxes in this particular fen is not dependent on the usual environmental controlling factors (water level or soil temperature, but see lines 408–410 about the role of deeper soil temperature).

Fourthly, we think those traits of *C. rostrata* (i.e., deep rooting system, partially evergreen nature) that could affect the flux rate are already stated in the text. We write in lines 359 onwards e.g. the following: "High amounts of vascular plant, sedge, and *C. rostrata biomass likely enhance methane production and release by supplying labile organic carbon substrates for methanogenesis through deep root systems throughout the year (Alm et al., 1999; Joabsson et al., 1999, Saarinen, 1996). High flux rates from C. rostrata dominated plots* (Fig. 7) may be due to the species' high methane transport rate (Ge et al., 2023), the high porosity of its roots (Ge et al., 2023), and a low capacity to oxidize methane into CO2 in the rhizosphere (Ström et al., 2005). Additionally, the perennial nature and deep rooting traits of *C. rostrata* (Saarinen, 1996; 1998) could support methane production and transport during the cold season by providing substrates for microbial processes in deeper peat layers and a potential pathway from belowground to the atmosphere. Approximately 40 % of *C. rostrata shoots at our study site* overwinter green (Cunow et al., unpublished data), indicating the potential to transport gases also during wintertime."

As two of the referees commented the wording of the title, we have chosen to slightly change the title and replace the word "controls" to the word "explains".

2. The second main criticism is regarding the visualization of the data. In my opinion the visualizations of the multivariate analyses should be included in the manuscript as it would underline the descriptive nature of the resuls section and make it more easy to follow.

As you are talking about spatial variation I think it would add to the value of the manuscript if you include a visualization of the spatial variation (perhaps a map of mean fluxes, or divided by season, one for snow season and one for snow-free season, size or colour as indicator for methane flux). Thereby it would be much easier to follow the overall story.

Response: We understand that lifting some figures from the appendix to the manuscript might help following the text and have now done so. When choosing locations for the figures, we were concerned about overloading the manuscript and therefore decided to place CCA graphs to the appendix instead of the manuscript.

We have added two supplementary figures to appendix B (Figs. B7, B8) to visualize the spatial variation of the mean fluxes in snow-free and snow seasons. We refer to these figures in lines 206 and 212).

**3. Just for consideration:**

As a large part of the variability of methane fluxes is unexplained you could consider using linear mixed effect models as your title suggests that methane is the response variable and you could include species as a random effect and results would make it statistically more robust as to which effect the plants have. It would be interesting to know how much of the variability can actually be explained.

Response: We thank you for this suggestion and agree that it could be an interesting analysis. However, at this stage, we decide not to perform any new analyses for this manuscript.

Generally, I find this work important especially under the consideration of the very strong data set and the importance of winter time flux measurements in order to assess regional if not global uncertainty.

Therefore, after the assessment of the above mentioned general concerns and below mentioned minor aspects by the authors, I would recomment to consider it further for publication.

Minor aspects:

55: have a "stronger" focus

Response: corrected

**Comment:**

High pH values  $\rightarrow$  it would be nice if you discuss the conditions of this fen in comparison to other northern rich fens shortly in the discussion to put the boundary conditions into perspective.

Response: Thank you for this observation. Although our study site exhibits relatively high pH values, rich fens in the northern boreal region do not universally share this characteristic. Groundwater pH in such habitats is often slightly below 7 (Hájek et al., 2021), reflecting regional differences in bedrock chemistry and hydrological inputs. For example, Olefeldt et al. (2017) describe a rich fen in interior Alaska with surface water pH of 5.2–5.4, yet still classified as rich due to its minerotrophic status and vegetation dominated by brown mosses and sedges. These examples highlight that rich fen classification depends more on hydrological connectivity, nutrient availability, and species composition than on absolute pH levels.

Our site's pH values (~7) are at the upper end of the range typically observed in northern boreal rich fens. For example, Laitinen et al. (2021) reported water pH values ranging from 4.0 to just over 6.4 across 18 eastern Finnish sloping fens. This suggests that while our site clearly qualifies as a rich fen, it may represent an unusually base-rich example within the northern boreal context.

We have added a short description of the boundary conditions to "2.1 Study site" in lines 76-77: "These values are relatively high and place our site at the upper end of the pH range typically observed in northern boreal rich fens, which often exhibit pH values below 7 (e.g., Hájek et al., 2021; Olefeldt et al., 2017; Laitinen et al., 2021)."

Likewise I find it necessary to at least also compare the measured fluxes with flux values from other comparable studies. To date I only found comparisons to as how large the contribution of e.g. the winter season fluxes is to the total annual flux (Alm et al.).

Response: In the previous manuscript version, we included a reference about flux rates from northern boreal rich fens (Jammet et al. 2017, see line 314). We chose to refer to this study as it resembles our study site. It should be noted that there are not too many studies published from northern boreal rich fens where methane fluxes would have been measured similarly to our methods.

77: WTD with one milimieter precision seems overly accurate.

Response: We agree with the referee and have corrected the WTD values to one centimeter accuracy.

84: half of the area was fenced → discuss if there were any differences?

Response: Added in lines 89–90: "--- and the differences in vegetation and methane fluxes between inside and outside of the fence were likely related to the hydrological conditions, rather than the effects of the exclusion (Väisänen et al., unpublished data)."

Suggestion: change the units of flux to scientific notation

Response: We have considered the units and choose to keep them as they were.

consider also using flux measurements with lower R2, what is with very low fluxes?

Response: We have considered them, as explained in lines 115-116: "We accepted the measurements with an  $R^2$  value  $\geq 0.95$  (n = 3589) and inspected all the rest (n = 691) individually, leaving out measurements showing very strong non-linearity or any other sign of failed measurement (n = 159)."

119: assuming an average flux over all days sounds like it could be prone to large overestimation - what about just linearly interpolating the data

Response: We do not believe this is a significant concern in our study, as the flux measurements were conducted at high frequency, with only a few longer gaps. Given that our analysis spans a full year, we expect that any potential under- or overestimation errors, depending on whether the preceding day had unusually high or low fluxes, are likely to be balanced out over the course of the year.

Moreover, if the referee's concern relates to diel variation in fluxes, we would like to emphasize that such variation is minimal at our study site, as discussed in lines 124-125 and in our earlier response. Therefore, we consider the use of average daily fluxes to be a reasonable and robust approach in this context.

127: using the biomass to determine the community composition? Does this go together? Biomass depends on species composition but you cannot determine community composition with biomass. Please explain more clearly.

Response: We determined the community composition by calculating the number of shoots for vascular plants and by estimating a percentage coverage for bryophytes (within the collar), which is explained in detail in section 2.4.

133: I do not quite get the marker. Was it a square within which you sampled the plant material?

Response: As stated in the text from line 137 onward: "...we first selected an area where vegetation heights resembled the heights of the vegetation within the collars. Then, we randomly threw a marker and selected the first ten non-flowering individuals of target vascular plant species close to the marker." We did not collect the samples within any particular area but selected the first individuals that could be found near the marker, which in our case was a hat that had randomly been thrown in a particular area where the vegetation was similar to the study site.

134: what is close? Sorry if this goes too much into details, but I do not quite get the randomised sampling.

Response: The sampling was done approximately 50-150 meters from the plots. We have edited the sentence in line 132 to clearly explain this.

134: where were the flowering individuals sampled?

Response: The flowering individuals were sampled similarly to the non-flowering ones. In other words, when we were collecting samples for a species which we needed also flowering shoots from, we sampled both non-flowering and flowering individual simultaneously.

135: see my main criticism: what about the below-ground biomass?

Response: See the response earlier.

153: why did you calculate a ratio? Please explain already here.

Response: We calculated this ratio as it could be a potential parameter in methane modeling. We added an explanation in lines 158–159.

173: just for clarity: you used BM for the calculation of difference (Bray-Curtis etc.)?

Response: Yes.

181: Here I think it would be great to have a visualization of DCA or CCA, especially the CCA in order to classify the potential bias in the data (plots) due to environmental variables. A visual approach would make the results section better understandable.

Response: The visualizations are now available in the main document.

Figure 3: This is a very nice plot. I would suggest to take a fix factor for the second y axis so that the numbers of the y axis are located at a certain horizontal line of the plot. This would make reading the plot easier. And still axis text could be somewhat larger.

Response: We thank the reviewer for this suggestion but, after trying, found that the temperature values as well as the axis title turned harder to read. Therefore, we chose to use the original figure format.

225: here again I would prefer a plot rather than the description.

Response: We provide cluster dendrograms and indicator values for the clusters in appendix B. The clusters can also be seen in CCA graph, which we moved to the manuscript, as well as on a map in the same appendix (Fig. B6)

241: just for clarity: first you are talking about the significant differences tested and visualized in figure 4, than you are talking about the correlation between the clusters and the fluxes in figure 4. To me testing differences between groups and correlation analyses are to different things and in the latter part you are not stating correlation coefficients but the F-statistic. Please make clear, what you did here or reword.

Response: Thank you for pointing this out. To clarify, all statistical tests in this paragraph are based on ANOVA and subsequent post hoc comparisons (see lines 248-250). To avoid confusion with correlation analyses, we have replaced the word "correlate" with "explain" or "show significant

effects," which more accurately reflect the methods used.

307-310: In my opinion this description is a bit tedious as it is hard to extract the central message. And the value of 6 mg seems a bit arbitrary. In principle we see this in boxplots. Consider rewording or deleting this part. Or perform and outlier test and see how many of those are C. Rostrata dominated. You point this out much clearer in 351.

Response: The point of this sentence is to highlight that high fluxes were also measured from plots, which were not grouped under C. rostrata-cluster in the clustering analysis but still had C. rostrata. The level of 6 mg  $CH_4/m^2/h$  was chosen as this level marks the highest level of fluxes (based on LOESS smoothed marginal means) seen in the full dataset (see figure 3). We have slightly edited the text for readability.

313: the root characteristics and belowground biomass should be more stressed here.

Response: We have added a reference to section 4.2 where this topic is discussed.

316: I would presume that the maxima during July and August are strong indications for a dependance on soil temperature and microbial activity or root activity. This is nicely discussed here.

Response: Thank you!

319: whose extent (?). Somehow reads strangely.

Response: Corrected in line 341.

338: consider: methane production and emission

Response: We think "fluxes" is better in this context, as that is what we have measured. We can only speculate on production, as we did not have any concrete data about it.

377: The peat layer: what is exactly meant with peat layer? As water table was mentioned before.

Response: By peat layer we mean the thickness of the peat layer from pear surface to mineral ground. We added "The thickness of.." to the start of the sentence in line 402.

384: as you state methane fluxes tend to correlate stronger with soil temperatures in deeper layers (25 cm).

What exactly do you mean with soil temperatures that connect to the water table? And as you measured the soil temperature at 5cm depth and state that these values did not sufficiently explain methane fluxes I think the statement that vegetation (based on this finding) is the prominent influential factor is a bit strong and should be relativised.

Response: We mean that peat temperature at 25 cm depth is more related to the water level depth (and water temperature) than the temperature of the air, whereas the peat temperature at 5 cm is linked to the air temperature. Since our study site is a flow-through fen with rather homogenous microtopography, water level and soil temperatures (at 5 cm depth), we believe that the temperature from 5 cm acts as a proxy for spatial variability of peat temperatures in deeper layers. This is, in fact, one of the main points of this paper to talk about species as an indicator/characteristic of a

community, and to show that this system is not dependent on the usual environmental controlling factors.

**Report #4**

This referee report concerns the revised manuscript. The initial submission was reviewed by two referees; their detailed comments, together with the author responses, have been taken into consideration by this referee, following evaluation of the revised manuscript at 'face value'.

This is a robust and important year-round study of the relationships between vegetation community composition and methane (CH4) fluxes in a boreal rich fen; an ecosystem type for which such data are very scarce. The spatiotemporal variability of CH4 fluxes has been analysed and interpreted based upon 4121 hard-won individual measurements, using a manual closed-chamber approach over 36 study plots year-round. The revised manuscript reads well, and this is an important contribution to the field, highlighting the potential to upscale emission predictions and improve ecosystem-scale CH4 modelling by identifying vegetation-related emission hotspots.

Response: We thank the reviewer for the insightful comments and appreciation for our work. Below we provide detailed responses for the comments.

The manuscript is very strong in its current form, although there are a few remaining aspects which might be worth further consideration to get the most out of this study:

On lines 116-20 the authors state that "Annual accumulated flux (1.11.2021- 31.10.2022) was estimated by calculating a 24-hour accumulated flux for each available datapoint by multiplying the hourly mean flux by 24. These daily flux values were then summed to obtain the annual total. The days which were missing a measurement were given the value from a previous measurement, assuming the fluxes did not vary remarkably diurnally or over the days". The assumption that 'fluxes did not vary remarkably diurnally' requires further justification/consideration, in my view (also; should 'diel' replace 'diurnal' – check definitions?). The measurements were taken between 8 am and 6 pm, using a clear polycarbonate chamber. Thus, photosynthesis and stomatal conductance will likely reflect daylight conditions, with open stomata (in living tissues, during the snow-free season). Is it possible, therefore, that extrapolation from day-time measurements to 24-hour flux values could cause a systematic overestimation of daily flux rates, where aerenchymatous CH4 transport is important (I have provided web-links to some potentially relevant papers at the end)? If release of CH4 via the stomatal pathway (as opposed to via leaf micropores and/or the epidermis/cuticle) is potentially important then I think it is worth noting in the manuscript. If no 'around-the-clock' flux measurements are available from this site then this does not undermine the paper; rather, this issue should be noted and discussed. Indeed, based on the results presented here it would be valuable, in any future study, to take some (snow-free season) 24-hour measurements, especially in plots belonging to the Carex rostrata cluster. Note that I am aware of the latitude of Puukkosuo fen (66.377299° N), and the implications for light climate.

Response: In our study site, two automatic chambers recorded fluxes continuously also during our study period. These data provide strong evidence that diel variation in  $CH_4$  fluxes is low at our site. Specifically, during the peak of the growing season and photosynthesis (July), when diel variation (of photosynthesis) is the most pronounced, the average daily standard deviation was 0.758 mg  $CH_4/m^2/h$  for Chamber 1 and 0.696 mg  $CH_4/m^2/h$  for Chamber 2. During snow season, the average daily standard deviation was 0.518 and 0.439 mg  $CH_4/m^2/h$ , for chambers 1 and 2, respectively.

Additionally, we have a set of manual 24-hour measurements from eight of the studied plots, which also support the conclusion of low diel variation (see figure below). We have clarified this in the manuscript (lines 124-125) and added a reference to the supporting data (Mastepanov et al., unpublished).

Put together, these findings suggest that our method of estimating daily fluxes by multiplying the hourly mean by 24 does not lead to systematic overestimation.

We also thank the reviewer for pointing out the terminology, and have replaced "diurnal" with "diel" throughout the manuscript for accuracy.

Lines 65-66, 336-37 and 384-85 (the final sentence of the Discussion) state, respectively, "We hypothesize that (1) the plant community composition affects the methane flux ...", and " ...plant functional type and species largely determine the magnitude of the fluxes" and "All these findings highlight that vegetation, rather than environmental factors, was the main driver of methane fluxes at our site." However, because plant community composition itself reflects (and interacts with) site physicochemical environmental factors, it is important to be very careful with the wording here, and assignment of 'cause and effect'. I would therefore urge the authors to reflect on this one more time, prior to final publication, and consider whether these statements remain robust and objective, or whether some caveats should be introduced. I am not disagreeing with these statements, but plant community composition is not independent of site-level environmental factors, which themselves may influence CH4 fluxes. Indeed, I wonder if the title of the paper could perhaps be amended (slightly!) to "Plant community composition explains spatial variation in year-round methane fluxes in a boreal rich fen"?

Response: We agree that plant community composition is not independent of environmental conditions. We observed that spatial variation in methane fluxes aligned closely with vegetation composition. While we acknowledge that vegetation can act as a proxy for underlying environmental factors, the conditions within our study site—such as microtopography, water level, and soil temperature at 5 cm depth—were relatively homogeneous. This supports our interpretation that vegetation was the primary driver of spatial variation in methane fluxes at this site.

This said, we recognize the possibility that unmeasured environmental variables may also contribute to the observed patterns. To reflect this nuance and maintain objectivity, we have slightly revised the title of the manuscript as suggested by the reviewer.

Related; lines 274-75 state that "There was no significant correlation between methane fluxes and WTD or soil temperature in any period." I found this remarkable, based on Fig. 3, which shows a broad relationship between soil temperature and CH4 fluxes for all vascular plant clusters on a seasonal basis. At the end of the Discussion section (lines 382-84), however, the authors explain that "methane fluxes did not correlate with peat temperature at 5 cm depth. Indeed, methane fluxes in boreal rich fens associate with deeper soil temperatures, which connect to water table position, rather than with surface temperatures influenced by air temperature (Olefeldt et al., 2017)." Had soil temperature data been available from deeper in the profile then do the authors consider that they might have been able to detect a relationship between temperature and CH4 flux; or is it solely, as they claim, that "vegetation, rather than environmental factors, was the main driver of methane fluxes at our site" (line 385)? Put another way, are the authors confident that this final statement, in the absence of relevant (deeper) soil temperature data, is robust?

Response: As noted in our response to Referee #3, we consider the 5 cm soil temperature to be a reasonable proxy for spatial variability in deeper peat layers, particularly in our study site, which is a flow-through fen with relatively homogeneous microtopography, water level, and soil temperature.

One of the central aims of our study is to highlight the role of vegetation composition as an indicator of methane flux potential, especially in a context where traditional environmental controls appear less variable. While Fig. 3 suggests a general relationship between soil temperature and methane fluxes across clusters, it is important to note that these are averaged values, and such patterns may not hold consistently at the level of individual plots. That said, we acknowledge the possibility that unmeasured environmental factors, such as deeper soil temperatures, could contribute to the observed flux patterns. Nonetheless, we believe our interpretation remains robust within the scope of the available data.

Some more minor points for consideration:

Lines 52-53 - A very bold statement appears here, reliant upon just one reference: "Climate change is predicted to accelerate the natural vegetational succession in boreal rich fens towards Sphagnum-dominated plant communities even in stable hydrological conditions (Kolari et al., 2021)." I would therefore suggest modifying the sentence to "Climate change is predicted to accelerate the natural, autogenic, vegetational succession in boreal rich fens towards Sphagnum-dominated plant communities, even in stable hydrological conditions (see Kolari et al. (2021), and references therein)."

Response: Thank you for pointing this out. We have corrected the reference as suggested in lines 53–54.

Line 135 – Delete the comma, to read " ... for those species which were found flowering ..."

Response: Corrected.

Line 150 – The units "g/1 %" appear, which in the manuscript font can look like g per litre. I therefore suggest writing this in full; i.e. g per 1%. Response: Good point. Corrected in line 154.

Caption of Figure B7 – correct the spelling of segregated (from segragated). Response: Corrected (now fig. B6).

References relating to CH4 transport through vascular plants:

https://doi.org/10.1104/pp.94.1.59

https://doi.org/10.1016/0045-6535(93)90430-D

https://doi.org/10.1016/0304-3770(93)90040-4

https://doi.org/10.1016/0304-3770(96)01048-0

https://doi.org/10.1016/j.atmosenv.2003.09.066

https://doi.org/10.1016/j.aquabot.2004.10.003

https://doi.org/10.1046/j.1469-8137.1998.00210.x

https://doi.org/10.1111/j.1469-8137.2012.04303.x

https://doi.org/10.1007/s10533-019-00600-6

https://doi.org/10.1002/lno.11467